# SCoA: Revisiting Domain Generalized Object Detection with Style-Conditioned Adaptation

Han Jiang[1]  Wenfei Yang[1,2]  Tianzhu Zhang[1,2]  Yongdong Zhang[1]

## Abstract

Domain generalized object detection (DGOD) aims to train an object detector on a single source domain and generalize it to unseen target domains. Recent advances in DGOD have increasingly exploited vision foundation models (VFMs) via parameter-efficient finetuning strategies. However, existing approaches typically adapt VFMs with fixed, style-agnostic parameters, overlooking that different visual styles may induce distinct task discrepancies. To address this challenge, we propose **SCoA**, a novel **S**tyle **Co**nditioned **A**daptation framework for dynamic, style-aware task compensation. Specifically, we introduce a Spectral Style Modeling (SSM) module that preserves local style cues via a memory-based mechanism, enabling diverse style characterization from a single source domain. Conditioned on the extracted style signals, we design a Mixture-of-Tokens Adaptation (MTA) mechanism, which maintains multiple adaptation tokens and dynamically routes each sample to an optimal combination of tokens, thereby explicitly modeling style-dependent task mismatches. In addition, we propose a Style-Conditioned Query Refinement (SCQR) module that injects style information into object queries, enabling a style-aware detection head. By jointly integrating these components, SCoA allows the model to follow style-specific adaptation trajectories, achieving effective and flexible task compensation for VFM-based DGOD. Extensive experiments demonstrate that the proposed SCoA achieves state-of-the-art performance across two challenging scenarios.

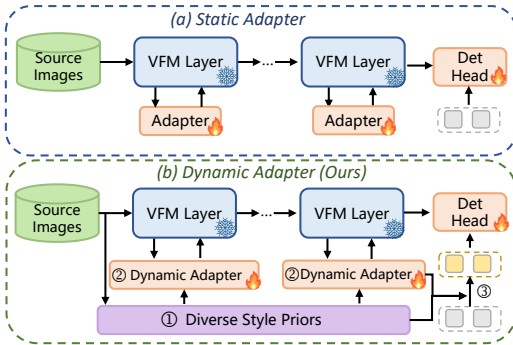

*Figure 1.* (a) Static adapters employ fixed parameters, performing uniform task compensation across all styles. (b) Our approach adopts a dynamic and style-aware adaptation paradigm, leveraging diverse style priors to condition both adapters and object queries.

## 1. Introduction

Object detection (Liu et al., 2016; Girshick, 2015; Ren et al., 2015) has achieved remarkable milestones in recent years. However, the success of modern detectors is heavily predicated on the independent and identically distributed (i.i.d.) assumption, where training and testing data are expected to be drawn from the same distribution. Consequently, deploying these models in real-world scenarios inevitably leads to performance degradation due to significant domain shifts, such as varying weather conditions and illumination changes. To mitigate this challenge, Domain Generalization for Object Detection (DGOD) (Lin et al., 2021; Vidit et al., 2023; Lu et al., 2022; Cheng et al., 2025a; Jiang et al., 2025b) has been proposed to transfer knowledge from a source domain to unseen target domains.

In recent years, Vision Foundation Models (VFMs), such as CLIP (Radford et al., 2021), DALL-E (Ramesh et al., 2021), DINOv2 (Oquab et al., 2024), and SAM (Kirillov et al., 2023), have demonstrated strong zero-shot capabilities through self-supervised pretraining on large-scale datasets. These properties make VFMs a promising backbone for DGOD. Nevertheless, directly adapting VFMs to object detection remains challenging due to a substantial task gap between pretraining objectives (e.g., contrastive learning or mask reconstruction) and the granular requirements of downstream dense prediction.

[1]School of Information Science and Technology, University of Science and Technology of China, Hefei, China. [2]National Key Laboratory of Deep Space Exploration, University of Science and Technology of China, Hefei, China. Correspondence to: Wenfei Yang <yangwf@ustc.edu.cn>.

*Proceedings of the 43rd International Conference on Machine Learning*, Seoul, South Korea. PMLR 306, 2026. Copyright 2026 by the author(s).

To address this challenge, most existing VFMs-based DGOD methods (Wei et al., 2024; Li et al., 2025) resort to parameter-efficient fine-tuning (PEFT) strategies (Jia et al., 2022; Wu et al., 2022), which introduce lightweight learnable adapters to bridge the gap between pretrained representations and downstream object detection tasks. Despite their efficiency, these adapters are optimized exclusively under the source-domain distribution, leading to a source-biased task compensation strategy. When deployed in unseen target domains with diverse visual styles, such biased task compensation manifests as style-dependent task mismatches, leading to degraded generalization performance. To alleviate this issue, several methods attempt to preserve style-invariant components of VFMs during fine-tuning. Specifically, style-related factors are suppressed either through causal intervention in the feature space (Li et al., 2025) or via singular value decomposition of model parameters (Yun et al., 2025). While effective in reducing domain-specific bias, such strategies inevitably compromise discriminability, resulting in poor confidence calibration and unreliable predictions (Yun et al., 2025). More importantly, the above approaches share a common limitation: they implicitly treat the discrepancy between VFMs pretraining and object detection as a static discrepancy across domains, as illustrated in Figure 1 (a) (Appendix E.). This assumption overlooks the fact that different styles may induce distinct task mismatches, which cannot be adequately addressed by a **static, style-agnostic** solution . Therefore, we naturally raise a fundamental question: *How can we effectively exploit the strong generalization capabilities of VFMs while dynamically adapting to diverse style conditions encountered in unseen domains?*

Driven by this question, we revisit VFMs adaptation for object detection and identify three key challenges from a dynamic and style-aware perspective, as shown in Figure 1 (b). ① Single-source datasets exhibit limited style diversity, making it difficult to synthesize diverse style priors that can effectively condition downstream adaptation. ② Designing dynamic adapters calls for a structured mechanism that jointly models shared task-level compensation and style-specific adaptations, enabling style-aware routing while preserving parameter efficiency and stable optimization. ③ Existing detection heads remain static across domains, preventing object queries from being aware of style-induced task discrepancies.

In this paper, we propose **SCoA**, a novel **S**tyle-**Co**nditioned **A**daptation framework, including a spectral style modeling (SSM) module, a mixture-of-tokens based adaptation mechanism (MTA), and a style-conditioned query refinement (SCQR) module to dynamically address style-aware task discrepancies in domain generalized object detection. Specifically, **to synthesize diverse style priors**, SSM models style information in the frequency domain by maintaining a mem-

ory that stores low-frequency amplitude components of local regions across the entire dataset. By adversarial weighting the memory entries, we generate challenging style signals, which are encoded into compact style vectors and used as the conditioning signals for downstream adaptation. **To enable style-conditioned task compensation**, we design a set of learnable tokens parameterized via low-rank decomposition, comprising a shared matrix that captures task-relevant commonalities and multiple style-specific matrices that model style-dependent adaptations. While the shared component is reused across inputs, the style-specific components are dynamically selected according to the extracted style vector. Furthermore, **to generate style-aware object queries**, in the SCQA, we fuse tokens from multiple backbone layers under the guidance of the style vector, and employ the aggregated tokens as semantic anchors to refine object queries in the detection head via cross-attention. This design establishes an explicit semantic linkage between tokens and detection queries, transforming static object queries into style-aware counterparts that can better accommodate style-induced task discrepancies. By integrating these components, SCoA enables the model to route samples toward optimal adaptation trajectories conditioned on input style, thereby achieving dynamic, style-conditioned task compensation for VFM-based object detection.

The major contributions of our work can be summarized as follows: 1) We propose SCoA, a style-conditioned adaptation framework that performs dynamic task compensation. To the best of our knowledge, this is the first work to reformulate the task gap between pre-training and dense detection as a dynamic, style-dependent discrepancy rather than a fixed source bias. 2) We design a unified framework consisting of three tightly coupled components, including a spectral style modeling module, a mixture-of-tokens adaptation mechanism, and a style-conditioned query refinement module. These components can be seamlessly integrated into existing VFMs in a trainable manner.3) Extensive experiments and ablation studies demonstrate the effectiveness of the proposed method, consistently outperforming state-of-the-art approaches across two challenging scenarios.

## 2. Related Work

### 2.1. Domain Generalized Object Detection

Domain Generalized Object Detection (DGOD) aims to enhance the generalization capability of object detectors to unseen domains. Existing DGOD approaches can be broadly categorized into two learning paradigms: learning from scratch (Wu & Deng, 2022; Liu et al., 2024b; Xu et al., 2023) and fine-tuning pretrained vision foundation models (Yun et al., 2025; Li et al., 2025). Early DGOD approaches predominantly follow the learning-from-scratch paradigm, where object detectors are fully trained using

only source-domain data. For example, CDSD (Wu & Deng, 2022) introduces a cyclic-disentangling module to extract domain-invariant representations from a single domain and incorporates a self-distillation module to improve detection performance on unseen target domains. AFDA (Danish et al., 2024) applies standard image corruptions to perturb input-level distributions, aligning predictions across various augmentations of a single image. However, the performance of these methods is inherently bounded by the limited semantic representation capability of classical backbones (e.g., ResNet50).

More recently, the emergence of vision foundation models (VFMs), pretrained on large-scale datasets via image-text or image-image contrastive learning, has introduced a new paradigm for DGOD. Most VFM-based approaches adopt frozen VFMs as backbone networks and apply parameter-efficient fine-tuning strategies to exploit their strong generalization ability while reducing training cost. For instance, SoMA (Yun et al., 2025) introduces singular value decomposed minor components adaptation to preserve generalizable components of VFMs. Cauvis (Li et al., 2025) employs cross-attention prompts to suppress spurious correlations and a dual-branch adapter to disentangle causal and spurious features. Despite their effectiveness, existing VFM-based DGOD methods generally assume a static task gap between pretraining objectives and downstream dense detection tasks, overlooking diverse style variations encountered in unseen environments. In contrast, our method utilizes diverse style information to perform dynamic, style-conditioned task compensation, enabling effective adaptation for VFMs.

## 2.2. Vision Foundation Models

Recently, Vision Foundation Models (VFMs) (Radford et al., 2021; Kirillov et al., 2023) have demonstrated strong zero-shot generalization capabilities through self-supervised pretraining on large-scale datasets. For instance, CLIP (Radford et al., 2021) learns high-quality visual representations through contrastive learning on large-scale image–text pairs. SAM (Kirillov et al., 2023) introduces a promptable segmentation framework pre-trained on a diverse and large-scale dataset, enabling flexible adaptation to various segmentation tasks. DINOv2 (Oquab et al., 2024) advances self-supervised visual pretraining by integrating objectives from prior works, resulting in strong performance across dense prediction benchmarks. The success of these models highlights the effectiveness of large-scale pretraining and makes VFMs attractive backbones for challenging scenarios such as domain generalization. Most existing efforts focus on segmentation tasks (Yi et al., 2024; Wei et al., 2024), where VFMs are typically fine-tuned using learnable tokens or lightweight adaptation modules. In contrast, the exploration of DGOD methods based on VFMs remains relatively limited.

## 3. Method

### 3.1. Preliminaries

**Problem Formulation.** In the context of domain generalized object detection, we have a labeled source domain and $T$ unseen target domains. Formally, we denote the source domain as $\mathcal{D}_s = \{(x_i, y_i)\}_{i=1}^{N_s}$, where $x_i \in \mathbb{R}^{H \times W \times 3}$ is an image and $y_i$ represents the corresponding detection annotations, including class labels and bounding boxes. $N_s$ is the number of training samples. The $T$ unseen target domains are represented as $\{\mathcal{D}_t\}_{t=1}^{T}$, whose data distributions differ from that of the source domain. The goal of DGOD is to learn a detection model based solely on $\mathcal{D}_s$ that achieves robust detection performance across all unseen target domains.

**Baseline Detector.** To achieve better generalization performance, we adopt frozen VFMs as the backbone of our detector. Specifically, given a pre-trained VFM with parameters $\theta_M$, which consists of a sequence of feature extraction layers $\{L_1, L_2, \ldots, L_N\}$, we introduce a lightweight adapter between layers within the backbone, which refines and forwards the feature maps from each layer to the subsequent one. On top of the adapted backbone, we employ a detection head $\mathcal{H}$ parameterized by $\theta_h$ for object localization and classification. The overall objective can be written as:

$$\arg\min_{\theta_r, \theta_h} \sum_{i=1}^{N_s} \mathcal{L}_{det}\big(\mathcal{H}_{\theta_h}\big(\mathcal{V}_{\theta_M, \theta_r}(x_i)\big), y_i\big), \qquad (1)$$

where $\mathcal{V}_{\theta_M, \theta_r}(\cdot)$ represents the forward process of the frozen VFM equipped with the adaptation strategy, and $\mathcal{L}_{det}(\cdot)$ denotes the standard object detection loss.

### 3.2. Overview

The overall architecture of our proposed SCoA is shown in Figure 2. The primary objective of our method is to perform a dynamic, style-conditioned task compensation for VFMs adaptation. Our approach consists of three key components: a spectral style modeling (Sec. 3.3), a mixture-of-tokens based adaptation (Sec. 3.4) and a style-conditioned query refinement (Sec. 3.5).

### 3.3. Spectral Style Modeling

In this section, we first transform the spatial features into frequency domain via Fast fourier transform (FFT) (Brigham, 1988), and contrust a style memory which covers diverse local low-frequency ampitude information of source data. Finally, we generate a new style mode via adversarial weighting style memory entries, which can be used as condition in the following adaptation and query refinement process.

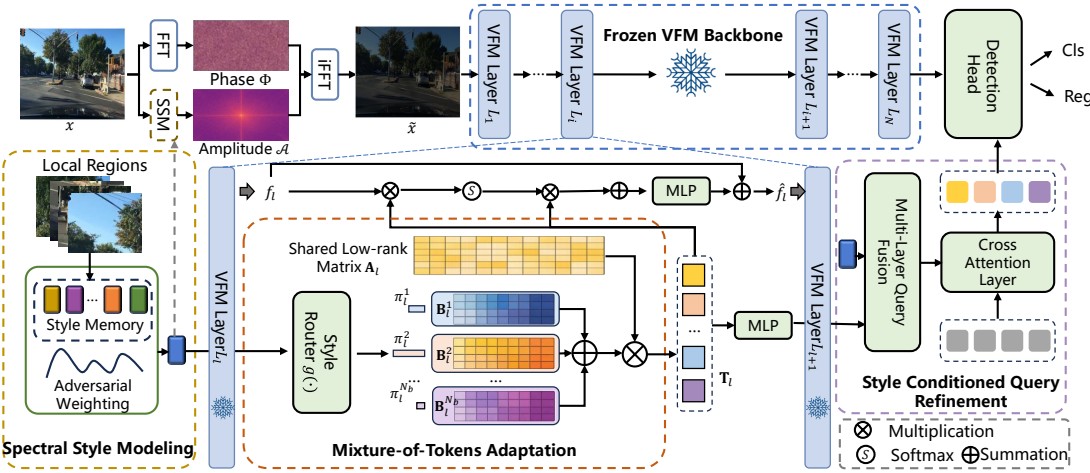

*Figure 2.* The overall architecture of our SCoA. Source images are first processed by the spectral style modeling module to generate augmented images along with their corresponding style vectors. The augmented images are then fed into VFMs for detection, while style vectors are used as conditioning signals to dynamically select optimal adaptation tokens. Finally, the learned multi-layer tokens are fused under the guidance of the style vectors and employed to refine object queries via a cross-attention mechanism, enabling the detection head to become explicitly aware of style variations.

**Frequency Representation.** Given an input image $x \in \mathbb{R}^{H \times W \times 3}$, we apply a 2D FFT to obtain its frequency representation:

$$\hat{x} = \mathcal{F}(x) = \mathcal{A}_x \odot e^{j\Phi_x}, \tag{2}$$

where $\mathcal{F}(\cdot)$ denotes the FFT operator, $\odot$ denotes element-wise product. $\mathcal{A}_x$ and $\Phi_x$ denote the amplitude and phase components, respectively. The low-frequency amplitude and high-frequency amplitude can be calculated with a pre-defined low-pass mask $\mathcal{M}_{lf}$:

$$\mathcal{A}_x^{lf} = \mathcal{M}_{lf} \odot \mathcal{A}_x, \quad \mathcal{A}_x^{hf} = (1 - \mathcal{M}_{lf}) \odot \mathcal{A}_x. \tag{3}$$

For subsequent computations, we define a spectral pooling operator that maps the low-frequency amplitude to a vector representation. Specifically, the operator extracts the central low-frequency block, resizes it to a fixed resolution, and then vectorizes the result:

$$\mathcal{P}_{\text{spec}}(A_x^{lf}) = \text{vec}\big(\mathcal{R}_l\big(\mathcal{C}_{h,w}(\mathcal{A}_x^{lf})\big)\big) \in \mathbb{R}^d, \quad d = 3l^2, \tag{4}$$

where $h = \frac{H}{2}, w = \frac{W}{2}$. The inverse operation, referred to as spectral unpooling, reconstructs the full-resolution spectrum by performing the inverse sequence of operations, including unvectorization, inverse resizing, and center embedding:

$$\mathcal{U}_{\text{spec}}(a) = \mathcal{E}_{h,w}\big(\mathcal{R}_{h,w}(\text{unvec}(a))\big), \qquad a \in \mathbb{R}^d. \tag{5}$$

This design enables efficient storage and manipulation of low-frequency style cues while preserving spatial alignment during reconstruction.

**Style Memory Construction.** Prior studies (Rao et al., 2023) have observed that local regions within an image often

exhibit significant style discrepancies, suggesting that a single image can serve as a rich style repository for image augmentation. Therefore, we maintain a learnable style memory $\mathcal{M}_s = \{a_k\}_{k=1}^K$, where each entry $a_k \in \mathbb{R}^d$ represents a prototypical low-frequency amplitude pattern. The memory captures diverse appearance modes aggregated from local regions across the source data. In practice, for each source image $x$, we randomly sample M crops $\{x_m\}_{m=1}^M$ at different spatial resolutions. For each crop, we achieve its low-frequency vector $a_m^{lf}$ by Eq. (3) and Eq. (4). We then compute the normalized similarity matrix between $a_m^{lf}$ and each memory entry $a_k$:

$$s_{k,m} = \frac{\exp(a_k \cdot a_m^{lf})}{\sum_{m=1}^M exp(a_k \cdot a_m^{lf}))}. \tag{6}$$

With the similarity matrix, the memory is updated by aggregating the low-frequency amplitudes from all crops via an exponential moving average (EMA):

$$a_k \leftarrow (1-\lambda)\,a_k \;+\; \lambda \frac{1}{M} \sum_{m=1}^M s_{k,m}\,a_m^{lf}, \tag{7}$$

where $\lambda$ is a predefined momentum coefficient. To prevent different memory entries from collapsing to similar style representations, we utilize a triplet loss on the style memory:

$$\mathcal{L}_{\text{tri}} = \sum_{k=1}^K \max\big(\gamma, \; \|a_k - a_k^+\|_2^2 - \|a_k - a_k^-\|_2^2 \big), \tag{8}$$

where $\gamma$ is a margin hyperparameter, $a_k^+$ and $a_k^-$ denotes the most similar and the second most similar vector with $a_k$ in $\{a_m^{lf}\}_{m=1}^M$, respectively.

**Adversarial Style Mixing.** Rather than randomly sampling from memory, we generate challenging style perturbations by adversarially reweighting the stored style components. Specifically, under the current model parameters, we seek a worst-case style mixture that maximizes the detection loss:

$$w^{\star} = \arg \max_{w \in \Delta^K} \mathcal{L}_{det}(f(\tilde{x}(w)), y), \qquad (9)$$

where $\Delta^K = \{w \mid w_k \geq 0, \sum_{k=1}^{K} w_k = 1\}$ and $\tilde{x}(w)$ denotes the style-perturbed image. We follow (Sagawa et al., 2019) to calculate the optimal $w^*$, and the detailed process is shown in Appendix F. The adversarial low-frequency vector is constructed as

$$a_{\mathrm{adv}} = \sum_{k=1}^{K} w_k^* a_k, \qquad (10)$$

which is subsequently fed into a style encoder $h(\cdot)$ to produce a style vector $s$. In addition, we use spectral unpooling in Eq.(5) to achieve $\mathcal{A}_{adv}$, which replaces the original low-frequency spectral component:

$$\hat{x}'(w) = \left(\mathcal{A}_x^{hf} + \mathcal{A}_{adv}\right) \odot e^{j\Phi_x}. \qquad (11)$$

The perturbed image is then obtained by inverse FFT:

$$\tilde{x}(w) = \mathcal{F}^{-1}(\hat{x}'(w)), \qquad (12)$$

which is subsequently fed into the vision foundation model for training.

### 3.4. Mixture-of-Tokens based Adaptation

Given the style vector $s$ produced by the spectral style modeling module, we perform dynamic task adaptation through a mixture-of-tokens mechanism. The core idea is to factorize task refinement into a shared component that captures common detection knowledge across styles, and a set of style-specific components that model style-dependent adaptations. These components are then dynamically weighted and composed according to the style information, enabling flexible and style-aware task compensation.

**Low-Rank Token Parameterization.** For each transformer layer $l$, we introduce $m$ learnable adaptation tokens $\mathbf{T}_l \in \mathbb{R}^{m \times c}$. To achieve parameter efficiency and disentangle shared and style-specific adaptation patterns, $\mathbf{T}_l$ is parameterized via a low-rank decomposition:

$$\mathbf{T}_l = \mathbf{A}_l \sum_{i=1}^{N_b} \pi_l^i \mathbf{B}_l^i, \qquad (13)$$

where $\mathbf{A}_l \in \mathbb{R}^{m \times r}$ is a shared projection matrix to capture task-aware adaptation directions, and $\mathbf{B}_l^i \in \mathbb{R}^{r \times c}$ encode style-specific variations. The mixture coefficients

$\pi^l = [\pi_l^1, \ldots, \pi_l^{N_b}]$ are predicted by a lightweight router conditioned on the style vector $s$. Specifically, the router first produces unnormalized routing logits:

$$z_l = g_l(s), \qquad (14)$$

where $g_l(\cdot)$ is a small MLP for the $l$-th layer. We then select the indices of the top-$k$ matrices with the highest logits, denoted as $\mathcal{I}_k^l$. The routing weights are computed by applying a softmax over the selected tokens:

$$\pi_l^i = \begin{cases} \dfrac{\exp(z_l^i)}{\sum_{j \in \mathcal{I}_k^l} \exp(z_l^j)}, & i \in \mathcal{I}_k^l, \\ 0, & \text{otherwise.} \end{cases} \qquad (15)$$

To prevent the style router from collapsing to a small subset of matrices, we introduce a balance regularization term $\mathcal{L}_{aux}$ on the routing distribution following (Fedus et al., 2022):

$$\mathcal{L}_{\mathrm{aux}} = \frac{N_b}{L} \sum_{l=1}^{L} \sum_{i=1}^{N_b} \mathbf{F}_l^i \, \mathbf{P}_l^i, \qquad (16)$$

where $\mathbf{P}_l^i$ and $\mathbf{F}_l^i$ denote the importance and usage frequency of matrix $\mathbf{B}_l^i$, respectively, as defined in (Fedus et al., 2022).

Given the features $f_l$ produced by the $l$-th layer, we employ the learnable tokens $T_l$ to guide the refinement of feature representations as (Wei et al., 2024). Specifically, we calculate feature-token similarity $\mathbf{S}_l = \mathrm{softmax}(\frac{f_l \times (\mathbf{T}_l)^T}{\sqrt{c}})$, where $c$ denotes the channel dimension. The final refined features can be calculated as:

$$\hat{f}_l = f_l + \left(\mathbf{S}_l \times (\mathbf{T}_l \times \mathbf{W}_{T_l} + \mathbf{b}_{T_l}) + f_l\right) \times \mathbf{W}_{f_l} + \mathbf{b}_{f_l}, \quad (17)$$

where $\mathbf{W}_{T_l}$ and $\mathbf{W}_{f_l}$ are learnable weight matrix, $\mathbf{b}_{T_l}$ and $\mathbf{b}_{f_l}$ are learnable biases. The refined features are passed to the subsequent layer. Note that except for learnable tokens $\mathbf{T}_l$ and style router $g_l(\cdot)$, all other parameters are shared across layers.

### 3.5. Style-Conditioned Query Refinement

While prior work (Wei et al., 2024) in segmentation directly employs learnable tokens as task queries, such a design is unsuitable for object detection, where queries are instance-specific and tightly coupled with bipartite matching. Replacing detection queries with generic task tokens would disrupt this instance-level correspondence. Therefore, we use the proposed tokens as style-conditioned modifiers to refine detection queries, rather than replacing them. Before query refinement, we design a multi-layer token fusion module to aggregate tokens across all layers.

Specifically, for each layer $l \in \{1, \ldots, N\}$, we first transform our leanable tokens $\mathbf{T}_l$ via linear transformation:

$$\hat{\mathbf{T}}_l = \mathbf{T}_l \times \mathbf{W}_t + \mathbf{b}_t, \quad \hat{\mathbf{T}}_l \in \mathbb{R}^{m \times c'}, \qquad (18)$$

*Table 1.* Domain generalization results on the Urban Scene dataset. Avg. denotes the average mAP across all out-of-domain scenarios.

| Methods | Params | DC | →DF | →DR | →NR | →NC | Avg. |
|---|---|---|---|---|---|---|---|
| *Backbone: ResNet101 (He et al., 2016)* | | | | | | | |
| FR (Ren et al., 2015) | 42.3M | 51.8 | 33.1 | 30.0 | 15.7 | 38.9 | 29.4 |
| S-DGOD (Wu & Deng, 2022) | 42.3M | 56.1 | 33.5 | 28.2 | 16.6 | 36.6 | 28.7 |
| CLIP-Gap (Vidit et al., 2023) | 42.3M | 51.3 | 38.5 | 32.3 | 18.7 | 36.9 | 31.6 |
| SRCD (Rao et al., 2023) | 42.3M | – | 35.9 | 28.8 | 17.0 | 36.7 | 29.6 |
| OA-DG (Lee et al., 2024) | 42.3M | 55.8 | 38.3 | 33.9 | 16.8 | 38.0 | 31.8 |
| PDOC (Li et al., 2024) | 42.3M | 53.6 | 39.1 | 33.7 | 19.2 | 38.5 | 32.6 |
| UFR (Liu et al., 2024b) | 42.3M | 58.6 | 39.6 | 33.2 | 19.2 | 40.8 | 33.2 |
| DivAlign (Danish et al., 2024) | 42.3M | 52.8 | 37.2 | 38.1 | 24.1 | 42.5 | 35.5 |
| SoMA (Yun et al., 2025) | 3.1M | 49.3 | 38.2 | 37.9 | 24.5 | 41.9 | 35.6 |
| FGT (He et al., 2025a) | 42.3M | - | 46.7 | 39.1 | 22.4 | 50.5 | 39.7 |
| *Backbone: DINOv2-L (Oquab et al., 2024)* | | | | | | | |
| Freeze | 0.0M | 65.0 | 46.9 | 55.0 | 42.8 | 54.2 | 49.7 |
| FFT | 307.3M | 68.2 | 47.2 | 56.6 | 43.1 | 57.1 | 51.0 |
| DoRA (Liu et al., 2024a) | 5.8M | 69.0 | 48.9 | 58.0 | 45.0 | 58.7 | 52.7 |
| AdaptFormer (Chen et al., 2022) | 6.3M | 68.9 | 49.8 | 58.3 | 44.4 | 58.8 | 52.8 |
| LoRA (Hu et al., 2022) | 5.5M | 69.6 | 49.5 | 58.1 | 46.1 | 59.6 | 53.3 |
| SoMA (Yun et al., 2025) | 4.9M | 69.4 | 51.0 | 59.3 | 47.6 | 59.3 | 54.3 |
| Cauvis (Li et al., 2025) | 4.7M | 73.7 | 56.5 | 64.6 | 47.6 | 61.2 | 57.5 |
| SCoA (Ours) | 8.4M | 75.9 | 59.3 | 65.1 | 49.8 | 63.5 | 59.4 |

*Table 2.* Comparison of different fine-tuning methods on the Urban Scene dataset. Avg. denotes the average mAP across all out-of-domain scenarios.

| Backbone | Finetune Method | DC | DF | DR | NR | NC | Avg. |
|---|---|---|---|---|---|---|---|
| EVA02 (Large) | Freeze | 63.0 | 45.2 | 48.6 | 27.3 | 38.8 | 40.0 |
| | +Linear | 57.8 | 39.2 | 40.5 | 21.2 | 33.3 | 33.6 |
| | +VPT-Deep (Jia et al., 2022) | 66.5 | 47.5 | 52.6 | 29.9 | 47.1 | 44.2 |
| | +EVP (Wu et al., 2022) | 63.2 | 45.5 | 50.1 | 28.6 | 39.2 | 40.9 |
| | +AdaptFormer (Chen et al., 2022) | 68.1 | 48.7 | 53.4 | 32.9 | 48.7 | 45.9 |
| | +SPT-Deep (Wang et al., 2024) | 66.4 | 47.5 | 52.7 | 31.8 | 47.2 | 44.8 |
| | +Rein (Wei et al., 2024) | 68.3 | 49.2 | 54.8 | 32.2 | 48.1 | 46.1 |
| | +Cauvis (Li et al., 2025) | 69.7 | 50.2 | 57.6 | 34.2 | 48.1 | 47.5 |
| | + SCoA (Ours) | 71.4 | 52.1 | 57.4 | 35.1 | 49.9 | 48.6 |
| SAM (Huge) | Freeze | 69.7 | 50.5 | 52.5 | 28.8 | 52.9 | 46.2 |
| | +Linear | 58.5 | 40.4 | 35.3 | 19.5 | 38.8 | 33.5 |
| | +VPT-Deep (Jia et al., 2022) | 63.7 | 46.0 | 44.9 | 24.8 | 45.2 | 40.2 |
| | +EVP (Wu et al., 2022) | 69.2 | 50.6 | 51.3 | 27.6 | 52.0 | 45.4 |
| | +AdaptFormer (Chen et al., 2022) | 70.7 | 52.7 | 55.1 | 31.8 | 54.4 | 48.5 |
| | +SPT-Deep (Wang et al., 2024) | 63.8 | 46.4 | 43.6 | 22.4 | 45.3 | 39.4 |
| | +Rein (Wei et al., 2024) | 70.0 | 51.9 | 54.0 | 30.9 | 54.4 | 47.8 |
| | +Cauvis (Li et al., 2025) | 72.2 | 53.7 | 55.8 | 31.9 | 55.7 | 49.3 |
| | + SCoA (Ours) | 74.9 | 55.8 | 56.8 | 34.2 | 56.3 | 50.8 |
| DINOv2 (Large) | Freeze | 71.2 | 53.5 | 60.8 | 42.6 | 59.5 | 54.1 |
| | +Linear | 55.7 | 35.8 | 31.8 | 18.8 | 35.3 | 30.4 |
| | +VPT-Deep (Jia et al., 2022) | 73.2 | 54.6 | 60.6 | 45.7 | 60.9 | 55.5 |
| | +EVP (Wu et al., 2022) | 71.9 | 55.7 | 60.7 | 48.4 | 59.4 | 56.1 |
| | +AdaptFormer (Chen et al., 2022) | 72.1 | 54.6 | 61.1 | 42.1 | 59.1 | 54.3 |
| | +SPT-Deep (Wang et al., 2024) | 73.2 | 55.7 | 62.6 | 46.6 | 60.6 | 56.4 |
| | +Rein (Wei et al., 2024) | 72.8 | 55.0 | 62.4 | 45.2 | 59.4 | 55.5 |
| | +Cauvis (Li et al., 2025) | 73.7 | 56.5 | 64.6 | 47.6 | 61.2 | 57.5 |
| | + SCoA (Ours) | 75.9 | 59.3 | 65.1 | 49.8 | 63.5 | 59.4 |

where $\mathbf{W}_t$ and $\mathbf{b}_t$ denote the weights and biases, respectively. $c'$ denotes the dimension of $\hat{\mathbf{T}}_l$. The style vector $s$ is then mapped to a set of style-conditioned fusion coefficients through a multi-layer projection:

$$\boldsymbol{\alpha} = \mathrm{Softmax}(\mathrm{MLP}_{\mathrm{sty}}(s)), \quad \boldsymbol{\alpha} \in \mathbb{R}^L. \quad (19)$$

Using these coefficients, we compute the final token representation by aggregating the transformed tokens:

$$\hat{\mathbf{T}} = \sum_{l=1}^{L} \alpha_l \hat{\mathbf{T}}_l. \quad (20)$$

The resulting token $\hat{\mathbf{T}}$ is used to refine the original DETR queries $\mathbf{Q} \in \mathbb{R}^{m_q \times c'}$ via a cross-attention operation. Specifically, queries are derived from $Q$, while keys and values are derived from the $\hat{\mathbf{T}}$:

$$\mathbf{Q} = \mathbf{Q}\mathbf{W}^{\mathcal{Q}}, \quad \mathbf{K} = \hat{\mathbf{T}}\mathbf{W}^{\mathcal{K}}, \quad \mathbf{V} = \hat{\mathbf{T}}\mathbf{W}^{\mathcal{V}}, \quad (21)$$

where $\mathbf{W}^{\mathcal{Q}}, \mathbf{W}^{\mathcal{K}}, \mathbf{W}^{\mathcal{V}} \in \mathbb{R}^{c' \times d_k}$ are learnable projection matrices. The style-conditioned query update is then obtained via the multi-head attention mechanism:

$$\hat{\mathbf{Q}} = \mathrm{softmax}\left(\frac{\mathbf{Q}\mathbf{K}^{\top}}{\sqrt{d_k}}\right)\mathbf{V}. \quad (22)$$

Finally, the refined queries are computed through a residual connection:

$$\tilde{\mathbf{Q}} = \mathbf{Q} + \hat{\mathbf{Q}}. \quad (23)$$

The refined queries $\tilde{\mathbf{Q}}$ are then fed into the decoder for object classification and localization. This refinement allows the detection head to adapt its classification and localization behavior to style-induced discrepancies while preserving instance-level query semantics.

### 3.6. Overall Loss

Following (Li et al., 2025), we employ DINO (Zhang et al., 2022) as our detection head, and the overall loss of our proposed SCoA is represented as follows:

$$\mathcal{L} = \mathcal{L}_{det} + \alpha\mathcal{L}_{tri} + \beta\mathcal{L}_{aux}, \quad (24)$$

where $\mathcal{L}_{det}$ is the supervised detection loss, $\alpha$ and $\beta$ are the hyperparameters used to balance the contribution of different losses.

## 4. Experiments

### 4.1. Experimental Setup

**Datasets.** Our experiments are primarily conducted on Urban Scene Dataset (Wu & Deng, 2022), which consists of driving scenes captured under five distinct weather conditions: Day-Clear (DC), Day-Foggy (DF), Dusk-Rainy (DR), Night-Rainy (NR), and Night-Clear (NC). The dataset covers seven object categories, including *person, car, bike, rider, motor, bus and truck*. For our experiments, we use Daytime Clear dataset as the source domain, which consists of 19,395 training images and 8,313 test images. The remaining four datasets are employed for testing, consisting of 26,158 images in Night Clear scene, 3,775 images in Daytime Foggy scene, 3,501 images in Dusk Rainy scene and 2,494 images in Night Rainy scene.

To evaluate robustness and generalization under distribution shifts, we further extend the evaluation to Cityscapes-C (Hendrycks & Dietterich, 2019), which contains 15 corruption types spanning four categories (noise, blur, weather,

*Table 3.* Robustness evaluation on Cityscapes-C. mPC is an average performance of 15 corruption types.

| Methods | Norm | Cityscapes→ Noise | | | Cityscapes→ Blur | | | | Cityscapes→ Weather | | | | Cityscapes→ Digital | | | | mPC |
| --- | --- | --- | --- | --- | --- | --- | --- | --- | --- | --- | --- | --- | --- | --- | --- | --- | --- |
| | | Gauss | Shot | Impul | Defoc | Glass | Motion | Zoom | Snow | Frost | Fog | Bright | Contr | Elas | Pixel | JPEG | |
| *Backbone: ResNet50 (He et al., 2016)* | | | | | | | | | | | | | | | | | |
| FR (Ren et al., 2015) | 42.2 | 0.5 | 1.1 | 1.1 | 17.2 | 16.5 | 18.3 | 2.1 | 2.2 | 12.3 | 29.8 | 32.0 | 24.1 | 40.1 | 18.7 | 15.1 | 15.4 |
| AutoAug (Zoph et al., 2020) | 42.4 | 0.9 | 1.6 | 0.9 | 16.8 | 14.4 | 18.9 | 2.0 | 1.9 | 16.0 | 32.9 | 35.2 | 26.3 | 39.4 | 17.9 | 11.6 | 15.8 |
| AugMix (Hendrycks et al., 2019) | 39.5 | 5.0 | 6.8 | 5.1 | 18.3 | 18.1 | 19.3 | 6.2 | 5.0 | 20.5 | 31.2 | 33.7 | 25.6 | 37.4 | 20.3 | 19.6 | 18.1 |
| SupCon (Khosla et al., 2020) | 43.2 | 7.0 | 9.5 | 7.4 | 22.6 | 20.2 | 22.3 | 4.3 | 5.3 | 23.0 | 37.3 | 38.9 | 31.6 | 40.1 | 24.0 | 20.1 | 20.9 |
| OA-DG (Lee et al., 2024) | 43.4 | 8.2 | 10.6 | 8.4 | 24.6 | 20.5 | 22.3 | 4.8 | 6.1 | 25.0 | 38.4 | 39.7 | 32.8 | 40.2 | 23.8 | 22.0 | 21.8 |
| GDD (He et al., 2025b) | 42.1 | 11.0 | 13.6 | 10.8 | 25.0 | 14.2 | 21.4 | 3.4 | 5.4 | 24.0 | 39.6 | 40.3 | 36.3 | 39.2 | 18.9 | 16.0 | 21.3 |
| FGT (He et al., 2025a) | 41.5 | 13.9 | 16.6 | 13.5 | 24.8 | 14.2 | 22.0 | 3.7 | 5.9 | 24.7 | 40.0 | 40.4 | 35.9 | 39.5 | 18.9 | 17.1 | 22.1 |
| *Backbone: DINOv2-L (Oquab et al., 2024)* | | | | | | | | | | | | | | | | | |
| Cauvis (Li et al., 2025) | 54.6 | 16.8 | 19.8 | 15.2 | 41.4 | 34.0 | 39.2 | 15.8 | 29.8 | 36.7 | 48.8 | 53.0 | 49.5 | 52.0 | 43.9 | 38.8 | 35.6 |
| DINOv2 (Oquab et al., 2024) | 48.2 | 16.9 | 18.8 | 13.1 | 40.7 | 25.9 | 39.7 | 15.4 | 28.2 | 32.1 | 40.5 | 48.4 | 46.7 | 48.5 | 39.8 | 30.4 | 32.3 |
| SCoA (Ours) | **60.9** | **23.4** | **23.9** | **20.6** | **47.1** | **36.7** | **45.6** | **19.3** | **36.2** | **40.1** | **54.5** | **58.7** | **56.1** | **55.7** | **46.2** | **41.9** | **40.4** |

and digital), each evaluated at five severity levels. All corruption patterns used in Cityscapes-C (e.g., motion blur, snow) are strictly excluded from the training data to ensure a fair evaluation.

**Implementation Details.** We implement our method based on the MMDetection framework. For optimization, we employ the AdamW optimizer with an initial learning rate of $1 \times 10^{-4}$ and a weight decay of $1 \times 10^{-4}$. The model is trained for 12 epochs with a total batch size of 8. All experiments are conducted on 4 NVIDIA RTX 3090 GPUs. Unless otherwise specified, the hyperparameters are set as $\alpha = \beta = 0.01$, $\gamma = 0.1$, $K = 64$, $N_b = 8$, $r = 16$ and $k = 3$. We report the mean average precision (mAP) at an Intersection over Union (IoU) threshold of 0.5. For Cityscapes-C, we further report the mean Performance under Corruption (mPC) (Hendrycks & Dietterich, 2019) as the robustness metric.

### 4.2. Comparison with State-of-the-arts Methods

In this section, we present a comparative analysis of our results against other state-of-the-art methods, including both ResNet-based methods and VFMs-based methods.

**Results on Urban Scene Datasets.** Table 1 shows the results on different weather conditions. Compared with prior DGOD methods built upon both conventional backbones (e.g., ResNet-101) and VFMs backbones (e.g., DinoV2-L), our approach achieves consistently superior performance across all adverse weather conditions. Specifically, compared with Cauvis, SCoA improves the average mAP by 1.9 % mAP, highlighting the effectiveness of modeling dynamic, style-conditioned task compensation. In addition, we compare our method with representative PEFT approaches across three vision foundation models, including EVA02 (Fang et al., 2024), SAM (Kirillov et al., 2023), and DINOv2 (Oquab et al., 2024). As shown in Table 2, our SCoA achieves 2.5 %, 3.0 % and 3.9 % mAP improvements over

*Table 4.* Ablation studies on Urban Scene Datasets..

| Setting | DF | DR | NR | NC | Avg. |
| --- | --- | --- | --- | --- | --- |
| *Incremental Build-up* | | | | | |
| Baseline | 54.7 | 62.6 | 44.9 | 59.1 | 55.3 |
| + SSM | 55.3 | 62.6 | 45.2 | 59.6 | 55.7 |
| + MTA | 58.4 | 64.8 | 49.1 | 62.6 | 58.7 |
| + SCQR (Full Model) | **59.3** | **65.1** | **49.8** | **63.5** | **59.4** |
| *Ablation on SSM* | | | | | |
| w/o Adv. Sampling | 59.0 | 64.1 | 49.2 | 63.1 | 58.9 |
| w/o Memory | 58.8 | 64.4 | 48.9 | 62.7 | 58.7 |
| *Ablation on MTA* | | | | | |
| w/o Shared $A$ | 58.2 | 64.3 | 49.1 | 62.2 | 58.5 |
| w/o LoRA | 58.4 | 64.5 | 49.7 | 62.9 | 58.9 |
| w/o Router | 56.1 | 62.7 | 46.1 | 60.7 | 56.4 |
| *Ablation on SCQR* | | | | | |
| w/o Fusion | 58.7 | 64.8 | 49.4 | 63.0 | 59.0 |
| w/o Style Condition | 58.9 | 64.8 | 49.5 | 63.1 | 59.1 |
| w/o cross attention | 55.9 | 61.3 | 43.2 | 58.5 | 54.7 |

Reins on EVA02, SAM, and DINOv2, respectively. These consistent gains across different VFMs demonstrate that SCoA can be seamlessly integrated into existing VFMs and provides an effective adaptation strategy.

**Results on Common Corruptions.** To evaluate robustness under diverse image corruptions, we conduct experiments on the Cityscapes-C benchmark. As reported in Table 3, our method achieves an mPC of 40.4% across all 15 corruption types, surpassing the backbone detector by a large margin of 8.1 % mPC. This result demonstrates that SCoA substantially enhances model robustness and generalization under out-of-distribution corruptions, rather than merely relying on the strong prior knowledge encoded in vision foundation models.

### 4.3. Quantitative Analysis

**Ablation Studies.** We conduct comprehensive ablation studies to analyze the effectiveness of each component in our framework, as summarized in Table 4. Starting from the baseline with frozen VFMs and token-based adapters, introducing SSM only brings a marginal improvement of 0.4% mAP, indicating that naive style augmentation provides limited additional gains for well-pretrained VFMs. In

*Table 5.* Effect of spectral style modeling.

| Method | DF | DR | NR | NC | Avg. |
|---|---|---|---|---|---|
| Image Corruptions (Danish et al., 2024) | 57.3 | 63.5 | 48.9 | 62.6 | 58.1 |
| PhysAug (Xu et al., 2025) | 57.9 | 64.2 | 48.5 | 62.8 | 58.4 |
| AdaIN (Huang & Belongie, 2017) | 58.8 | 64.7 | 49.5 | 63.0 | 59.0 |
| **Ours** | **59.3** | **65.1** | **49.8** | **63.5** | **59.4** |

*Table 6.* Effect of mixture-of-tokens based adaptation. **T** and **S** are drived from Eq (17).

| Method | DF | DR | NR | NC | Avg. |
|---|---|---|---|---|---|
| w/o MTA | 56.5 | 62.7 | 47.0 | 61.4 | 56.9 |
| Shuffle | 52.1 | 56.3 | 43.9 | 49.2 | 50.4 |
| Token Modulation ($\mathbf{T} \odot g(s)$) | 57.7 | 64.1 | 48.2 | 62.4 | 58.1 |
| Attention Modulation ($\mathbf{S} \odot g(s)$) | 58.0 | 63.4 | 48.9 | 62.8 | 58.3 |
| **Ours** | **59.3** | **65.1** | **49.8** | **63.5** | **59.4** |

contrast, incorporating style information via MTA brings the most significant improvement of 3.0% mAP, highlighting the importance of dynamic style-conditioned adaptation. Adding SCQR further improves the average mAP to 59.4, demonstrating that learnable tokens can further benefit detection performance. We then analyze each module in detail. For SSM, replacing adversarial style sampling with random sampling results in a 0.5% mAP drop, while removing the style memory causes a larger degradation of 0.7% mAP, indicating that diverse source-domain style information is critical for improving adapter generalization to unseen domains. Within the MTA framework, removing the shared matrix **A** forces all decomposed token parameters to become style-dependent, which leads to a noticeable performance degradation. This observation indicates that explicitly decoupling task-relevant commonality from style-specific variations is crucial for effective and robust adaptation. Notably, replacing the style-conditioned router with a random alternative causes a significant performance drop (59.4% vs. 56.4 %), which verifies that the gains of SCoA do not stem from increased model capacity but from effective dynamic adaptation. Finally, we analyze the SCQR module. Using only the final-layer tokens results in a 0.4% mAP reduction, while uniformly weighting tokens across layers leads to a 0.3% mAP drop, demonstrating the importance of adaptive cross-layer token aggregation. Moreover, directly replacing query refinement with learnable tokens causes a substantial 4.7 % mAP drop by disrupting instance-level query semantics and Hungarian matching.

**Effect of Spectral Style Modeling.** The proposed spectral style modeling aims to generate diverse style signals, which are then used as conditions for the subsequent dynamic adaptation. To prove the effectiveness of the proposed module, we explore different data augmentation methods, as shown in Table 5. Compared with common image augmentation methods (Xu et al., 2025; Danish et al., 2024) that rely on predefined perturbation strategies, our proposed

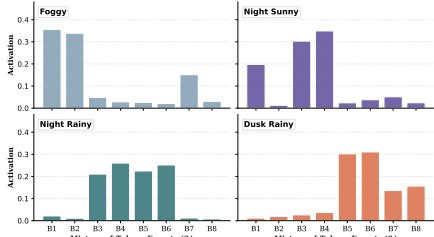

*Figure 3.* Visualization of routing patterns across unseen domains.

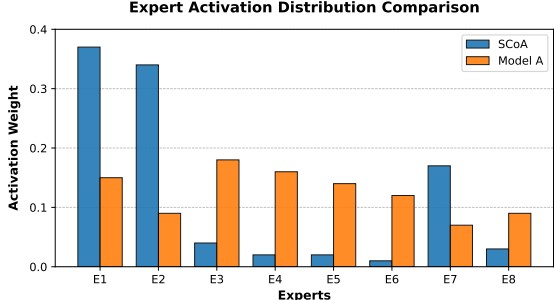

*Figure 4.* Comparison of routing weights between Model A and our SCoA.

method leads to at least 1.0 % performance gain. This indicates that maintaining local region styles in a memory-based manner provides more diverse and representative style information. In addition, augmenting images in the RGB space using AdaIN results in a 0.4% mAP degradation compared to frequency-domain modeling, further demonstrating the advantage of spectral representations.

**Effect of Mixture-of-Tokens based Adaptation.** Table 6 demonstrates the effectiveness of the proposed MTA. We first evaluate a shuffled variant by randomly permuting the style routing during inference. This results in a significant performance degradation, suggesting that the observed performance gains primarily stem from meaningful, style-aware task compensation. We further compare two simplified style-conditioned variants that employ either token-level modulation or attention-level modulation. Both variants yield degraded performance, with 1.3 % and 1.1 % mAP drops compared to our full model, respectively. These results highlight the necessity of the proposed mixture-of-tokens design for effectively modeling style-dependent task discrepancies. Furthermore, Figure 3 provides further insights into the behavior of the proposed mixture-of-tokens mechanism. We observe that different unseen domains activate distinct token distributions, demonstrating that the proposed MTA can dynamically route style information through appropriate token combinations.

**Discussion.** We investigate the scenario where the style memory does not adequately cover a target style. We observe that DF is clearly separated from the other three do-

mains (See in Figure 10 in Appendix). Based on this observation, we construct an experiment in which the style memory is populated only with low-frequency statistics from DR/NR/NC, thereby simulating a missing coverage of DF-style patterns (denoted as **Model A**). As a result, the performance on DF drops by 3.4% (55.9% vs. 59.3% ). To further analyze this effect, we visualize the routing weights in Fig. 4. We find that the routing distribution becomes significantly smoother, indicating that the router loses its discriminative, style-specific behavior and instead degenerates into a static PEFT mechanism.

## 5. Conclusion

In this paper, we propose SCoA, a novel style conditioned adaptation framework to perform dynamic, style-aware task compensation for VFMs-based domain generalized object detection. SCoA consists of three key modules: a spectral style modeling that synthesizes challenging and diverse style priors, a mixture-of-tokens based adaptation mechanism that performs style-conditioned task compensation, and a style-conditioned query refinement module that enables object queries to be aware of style variants. Extensive experiments demonstrate that our method significantly outperforms existing approaches across two challenging domain generalization datasets.

## Acknowledgements

This work was supported by the Science and Application Research Project of the Tianwen-2 Mission for Planetary Exploration Engineering under Grant TW2-01-001.

## Impact Statement

This paper presents work whose goal is to advance the field of machine learning. There are many potential societal consequences of our work, none of which we feel must be specifically highlighted here.

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

# A. Overview

In the supplementary material, we first show class-wise results on Urban Scene dataset and Real to Artist dataset. (Sec. B). We then conduct additional ablation studies (Sec. C) and extend our method with Faster RCNN detection head to further validate the effectiveness of the proposed method (Sec. D). Consequently, we analysis connections between style and task mismatch ((Sec. E). We show algorithm details (Sec. F) and present more detailed analysis of our method (Sec. G). Finally, we provide more visualization of style memory diversity, failure cases and detection results (Sec. H).

# B. The Class-wise Results

**Results on Daytime Clear Scene.** Table 7 reports the performance on the source-domain Day Clear setting. Our method achieves the best result of 75.9% mAP, outperforming other ResNet-based approaches by at least 17.3%, which demonstrates that vision foundation models (VFMs) provide a strong backbone for DGOD. Furthermore, our method surpasses Cauvis by 2.2% mAP, suggesting that the proposed dynamic, style-conditioned adapter does not sacrifice source-domain performance, but instead provides consistent gains.

**Results on Daytime Foggy Scene.** On the Daytime Foggy dataset, our method achieves the best overall performance with 59.3% mAP, consistently outperforming all competing approaches. Compared with the strongest baseline Cauvis, SCoA yields a +2.7% mAP improvement. Notably, our gains are particularly pronounced on object categories such as Car, Person, and Rider, which are known to be highly sensitive to visibility degradation caused by fog. This improvement indicates that the proposed style-conditioned adaptation can effectively compensate for fog-induced task discrepancy, enabling more robust object localization and recognition.

**Results on Dusk Rainy Scene.** For the Dusk Rainy scenario, which combines challenging illumination changes and rain-induced artifacts, SCoA attains the highest performance of 65.1 % mAP, surpassing UFR by 31.4 % mAP. We observe consistent improvements across most categories, especially Car and Person, demonstrating the robustness of our method under simultaneous illumination and weather shifts.

**Results on Night Rainy Scene.** On the Night Rainy dataset, which represents one of the most challenging unseen domains due to severe illumination degradation and rain noise, the proposed SCoA achieves the best 49.8 % mAP. In particular, our method shows competitive per-

*Table 7.* Class-wise quantitative results on Day Clear (source domain).

| Method | Bus | Bike | Car | Mot. | Per. | Rid. | Tru. | mAP |
|---|---|---|---|---|---|---|---|---|
| IterNorm (Huang et al., 2019) | 58.4 | 34.2 | 42.4 | 44.1 | 31.6 | 40.8 | 55.5 | 43.9 |
| IBN-Net (Pan et al., 2018) | 63.6 | 40.7 | 53.2 | 45.9 | 38.6 | 45.3 | 60.7 | 49.7 |
| SW (Pan et al., 2019) | 62.3 | 42.9 | 53.3 | 49.9 | 39.2 | 46.2 | 60.6 | 50.6 |
| ISW (Choi et al., 2021) | 62.9 | 44.6 | 53.5 | 49.2 | 39.9 | 48.3 | 60.9 | 51.3 |
| ClipGap (Vidit et al., 2023) | 55.0 | 47.8 | 67.5 | 46.7 | 49.4 | 46.7 | 54.7 | 52.5 |
| CDSD (Wu & Deng, 2022) | 68.8 | 50.9 | 53.9 | 56.2 | 41.8 | 52.4 | 68.7 | 56.1 |
| UFR (Liu et al., 2024b) | 66.8 | 51.0 | 70.6 | 55.8 | 49.8 | 48.5 | 67.4 | 58.6 |
| Cauvis(Li et al., 2025) | 76.3 | 64.3 | 87.7 | 64.2 | 79.2 | 72.1 | 71.8 | 73.7 |
| **SCoA(Ours)** | **79.2** | **65.9** | **90.5** | **67.1** | **79.7** | **74.3** | **74.9** | **75.9** |

formance on all categories, indicate that SCoA can learn more discriminative features even under extreme appearance changes.

**Results on Night Clear Scene.** For the Night Clear setting, SCoA obtains the best result of 63.5% mAP, exceeding ClipGap by 26.6% mAP. Despite the absence of rain, significant illumination differences between day and night still introduce substantial domain shifts. The consistent improvements across all categories indicate the effectiveness and robustness of our approach.

# C. More Ablation Studies

## C.1. Hyper-parameter Analysis

We conduct a comprehensive hyper-parameter sensitivity analysis to investigate the robustness of the proposed method, as illustrated in Figure 5.

**Effect of $\alpha$ and $\beta$.** We observe that the performance varies smoothly with respect to $\alpha$ and $\beta$. The mAP increases as $\alpha$ or $\beta$ grows from 0.001 to 0.01 and reaches the peak at $\alpha = \beta = 0.01$ with 59.4% mAP. Overall, the method is not overly sensitive to moderate changes in $\alpha$ and $\beta$.

**Effect of $\lambda$.** We further analyzes the impact of the EMA coefficient $\lambda$ used for updating the style memory. When $\lambda$ increases from 0.01 to 0.1, the performance consistently improves, indicating that a moderate update rate enables the memory to effectively evolve new style statistics from input images while preserving historical stability. However, overly large $\lambda$ (e.g., 0.5) leads to a noticeable performance drop, suggesting that aggressive memory updates may cause unstable or noisy style representations. The best performance is achieved at $\lambda = 0.1$, which provides a favorable balance between adaptability and stability in memory evolution.

**Effect of style memory size $K$.** As shown in Figure 5, increasing $K$ from 16 to 64 consistently improves performance, as richer style priors become available for conditioning. Further enlarging the memory size yields no additional

*Table 8.* Class-wise quantitative results on unseen target domain datasets.

| Method | Daytime Foggy | | | | | | | | Dusk Rainy | | | | | | | |
|---|---|---|---|---|---|---|---|---|---|---|---|---|---|---|---|---|
| | Bus | Bike | Car | Mot. | Per. | Rid. | Tru. | mAP | Bus | Bike | Car | Mot. | Per. | Rid. | Tru. | mAP |
| FR (Ren et al., 2015) | 34.5 | 29.6 | 49.3 | 26.2 | 33.0 | 35.1 | 26.7 | 33.5 | 34.2 | 21.8 | 47.9 | 16.0 | 22.9 | 18.5 | 34.9 | 28.0 |
| SW (Pan et al., 2019) | 30.6 | 36.2 | 44.6 | 25.1 | 30.7 | 34.6 | 23.6 | 30.8 | 35.2 | 16.7 | 50.1 | 10.4 | 20.1 | 13.0 | 38.8 | 26.3 |
| IterNorm (Huang et al., 2019) | 29.7 | 21.8 | 42.4 | 24.4 | 26.0 | 33.3 | 21.6 | 28.4 | 32.9 | 14.1 | 38.9 | 11.0 | 15.5 | 11.6 | 35.7 | 22.8 |
| CDSD (Wu & Deng, 2022) | 32.9 | 28.0 | 48.8 | 29.8 | 32.5 | 38.2 | 24.1 | 33.5 | 37.1 | 19.6 | 50.9 | 13.4 | 19.7 | 16.3 | 40.7 | 28.2 |
| SRCD (Rao et al., 2023) | 36.4 | 30.1 | 52.4 | 31.3 | 33.4 | 40.1 | 27.7 | 35.9 | 39.5 | 21.4 | 50.6 | 11.9 | 20.1 | 17.6 | 40.5 | 28.8 |
| ClipGap (Vidit et al., 2023) | 36.2 | 34.2 | 57.9 | 34.0 | 38.7 | 43.8 | 25.1 | 38.5 | 37.8 | 22.8 | 60.7 | 16.8 | 26.8 | 18.7 | 42.4 | 32.3 |
| PDOC(Li et al., 2024) | 36.1 | 34.5 | 58.4 | 33.3 | 40.5 | 44.2 | 26.2 | 39.1 | 39.4 | 25.2 | 60.9 | 20.4 | 29.9 | 16.5 | 43.9 | 33.7 |
| UFR (Liu et al., 2024b) | 36.9 | 35.8 | 61.7 | 33.7 | 39.5 | 42.2 | 27.5 | 39.6 | 37.1 | 21.8 | 67.9 | 16.4 | 27.4 | 17.9 | 43.9 | 33.2 |
| Cauvis (Li et al., 2025) | 50.7 | **43.8** | 73.8 | 50.5 | 67.4 | 61.9 | 47.5 | 56.5 | 67.2 | **54.8** | 85.6 | 52.0 | 65.5 | 57.2 | **70.1** | 64.6 |
| **ScoA (Ours)** | **53.2** | 43.7 | **79.5** | **51.8** | **69.8** | **67.5** | **49.6** | **59.3** | **68.5** | 50.9 | **87.4** | **55.1** | **67.6** | 57.4 | 68.9 | **65.1** |

| Method | Night Rainy | | | | | | | | Night Clear | | | | | | | |
|---|---|---|---|---|---|---|---|---|---|---|---|---|---|---|---|---|
| | Bus | Bike | Car | Mot. | Per. | Rid. | Tru. | mAP | Bus | Bike | Car | Mot. | Per. | Rid. | Tru. | mAP |
| FR (Ren et al., 2015) | 21.3 | 7.7 | 28.8 | 6.1 | 8.9 | 10.3 | 16.0 | 14.2 | 43.5 | 31.2 | 49.8 | 17.5 | 36.3 | 29.2 | 43.1 | 35.8 |
| SW (Pan et al., 2019) | 22.3 | 7.8 | 27.6 | 0.2 | 10.3 | 10.0 | 17.7 | 13.7 | 38.7 | 29.2 | 49.8 | 16.6 | 31.5 | 28.0 | 40.2 | 33.4 |
| IterNorm (Huang et al., 2019) | 21.4 | 6.7 | 22.0 | 0.9 | 9.1 | 10.6 | 17.6 | 12.6 | 38.5 | 23.5 | 38.9 | 15.8 | 26.6 | 25.9 | 38.1 | 29.6 |
| CDSD (Wu & Deng, 2022) | 24.4 | 11.6 | 29.5 | 9.8 | 10.5 | 11.4 | 19.2 | 16.6 | 40.6 | 35.1 | 50.7 | 19.7 | 34.7 | 32.1 | 43.4 | 36.6 |
| SRCD (Rao et al., 2023) | 26.5 | 12.9 | 32.4 | 0.8 | 10.2 | 12.5 | 24.0 | 17.0 | 43.1 | 32.5 | 52.3 | 20.1 | 34.8 | 31.5 | 42.9 | 36.7 |
| ClipGap (Vidit et al., 2023) | 28.6 | 12.1 | 36.1 | 9.2 | 12.3 | 9.6 | 22.9 | 18.7 | 37.7 | 34.3 | 58.0 | 19.2 | 37.6 | 28.5 | 42.9 | 36.9 |
| PDOC (Li et al., 2024) | 25.6 | 12.1 | 35.8 | 10.1 | 14.2 | 12.9 | 22.9 | 19.2 | 40.9 | 35.0 | 59.0 | 21.3 | 40.4 | 29.9 | 42.9 | 38.5 |
| UFR (Liu et al., 2024b) | 29.9 | 11.8 | 36.1 | 9.4 | 13.1 | 10.5 | 23.3 | 19.2 | 43.6 | 38.1 | 66.1 | 14.7 | 49.1 | 26.4 | 47.5 | 40.8 |
| Cauvis (Li et al., 2025) | 60.8 | 32.4 | 69.5 | **24.0** | 49.9 | 39.5 | **57.2** | 47.6 | 62.8 | 55.1 | 79.5 | **39.3** | **70.4** | 57.4 | 64.2 | 61.2 |
| **ScoA (Ours)** | **64.6** | **34.1** | **73.6** | 23.1 | **50.8** | **47.2** | 55.3 | **49.8** | **67.2** | **58.3** | **84.0** | 38.1 | 67.4 | **60.9** | **68.5** | **63.5** |

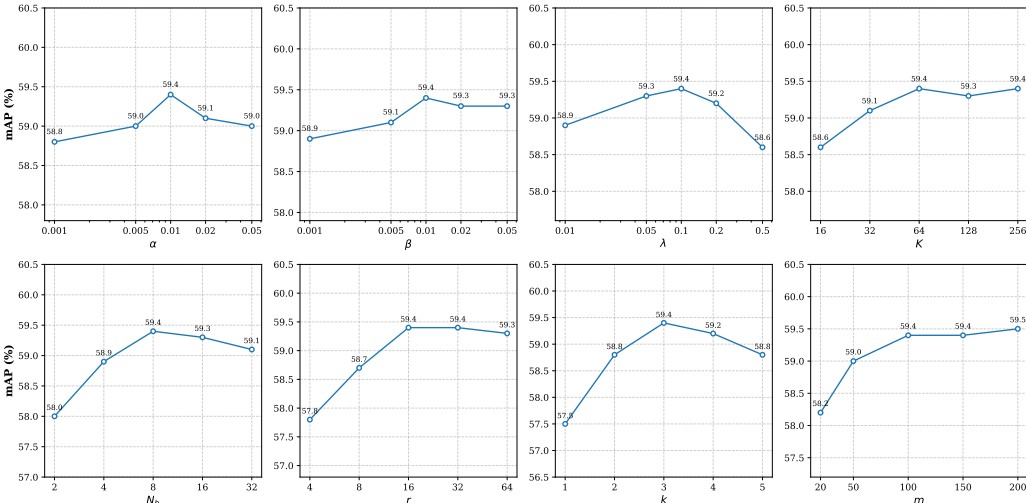

*Figure 5.* Hyper-parameters Analysis.

gains and even slightly degrades performance, indicating that redundant style representations may introduce noise.

provides a good balance between adaptation diversity and optimization stability.

**Effect of the number of experts $N_b$.** We study the influence of the total number of experts $N_b$ used in the mixture-of-tokens adaptation. As $N_b$ increases from 2 to 8, the performance consistently improves, indicating that a larger pool of experts enables more fine-grained and style-specific task compensation. However, further increasing $N_b$ yields marginal gains or slight degradation, suggesting that an excessive number of experts may introduce redundancy and complicate expert routing. Therefore, we set $N_b = 8$, which

**Effect of matrix decompose rank $r$.** For the matrix decompose rank $r$, the performance improves significantly when increasing $r$ from 4 to 16, highlighting the importance of sufficient adaptation capacity. Further increasing $r$ fails to capture additional informative factors, resulting in performance saturation or slight degradation.

**Effect of the top-$k$ expert selection $k$.** We investigate the impact of the number of selected experts $k$ in the top-

*Table 9.* Extend our method with Faster-RCNN head under common corruptions.

| Methods | Norm | Gauss | Shot | Impul | Defoc | Glass | Motion | Zoom | Snow | Frost | Fog | Bright | Contr | Elas | Pixel | JPEG | mPC |
|---|---|---|---|---|---|---|---|---|---|---|---|---|---|---|---|---|---|
| FR+DINO | 44.0 | 14.5 | 16.3 | 13.5 | 33.9 | 27.3 | 33.1 | 13.7 | 25.2 | 31.5 | 39.5 | 42.5 | 38.9 | 42.1 | 35.9 | 31.8 | 29.3 |
| Cauvis | 54.6 | 16.8 | 19.8 | 15.2 | 41.4 | **34.0** | 39.2 | 15.8 | 29.8 | 36.7 | 48.8 | 53.0 | 49.5 | 52.0 | 43.9 | 38.8 | 35.6 |
| **SCoA** | **57.3** | **18.2** | **21.5** | **16.9** | **45.8** | 32.1 | **40.5** | **16.2** | **34.7** | 36.9 | **49.0** | **56.7** | **53.1** | **52.3** | **45.7** | **39.0** | **38.5** |

$k$ routing mechanism. When $k$ increases from 1 to 3, the performance consistently improves, indicating that aggregating multiple experts provides complementary adaptation capabilities for handling diverse style-induced task discrepancies. However, further increasing $k$ leads to performance degradation, as involving too many experts may introduce irrelevant or conflicting adaptation signals, weakening the specialization of individual experts.

**Effect of the number of tokens** $m$**.** Finally, we observe that increasing $m$ improves performance up to $m = 100$, after which the gains saturate. This suggests that a sufficient number of tokens is required to model diverse adaptation patterns, while excessively large token sets offer limited benefits.

### C.2. Effect of different loss

Table 10 presents an ablation study on the memory diversity loss $\mathcal{L}_{div}$ and the load balancing loss $\mathcal{L}_{aux}$. When $\mathcal{L}_{div}$ is excluded, the performance decreases by 0.6% mAP compared to the full model, indicating that insufficient diversity in the style memory limits the coverage of style variations. Similarly, removing $\mathcal{L}_{aux}$ leads to a 0.5% mAP drop, suggesting that unbalanced expert utilization introduces routing bias and degrades adaptation stability. Notably, discarding both losses causes the largest degradation of 1.2% mAP, demonstrating that memory diversity and expert load balancing play complementary roles in our framework.

### D. Performance with Faster-RCNN

Table 9 reports results under common image corruptions. As shown in Table 9, since Cauvis is originally implemented with a Faster R-CNN detection head, we adopt the same Faster R-CNN head in our implementation of SCoA to ensure a fair comparison and eliminate potential performance bias introduced by different detection heads. Under this unified setting, SCoA consistently outperforms Cauvis across most corruption types and achieves 38.5 % mPC. This indicates that the robustness gains of SCoA are not attributed to architectural advantages of the detection head, but stem from its style-conditioned adaptation mechanism.

*Table 10.* Effect of diversity loss ($\mathcal{L}_{div}$) and balance regularization loss ($\mathcal{L}_{aux}$).

| $L_{\text{div}}$ | $L_{\text{aux}}$ | DF | DR | NR | NC | Avg. |
|---|---|---|---|---|---|---|
| | | 58.2 | 64.0 | 48.1 | 62.3 | 58.2 |
| ✓ | | 59.0 | 64.6 | 49.3 | 62.8 | 58.9 |
| | ✓ | 58.9 | 64.9 | 48.5 | 62.8 | 58.8 |
| ✓ | ✓ | **59.3** | **65.1** | **49.8** | **63.5** | **59.4** |

---

**Algorithm 1** Adversarial Style Mixing via GDPO (Sagawa et al., 2019) on the Simplex

---

**Input:** Image $x$, style memory $\{a_k\}_{k=1}^K$, step size $\eta$, steps $T$

**Output:** Adversarial image $\tilde{x}$

  Initialize $w^{(0)} \in \Delta^K$

  **for** $t = 0$ **to** $T - 1$ **do**

$$a_{\text{adv}}^{(t)} = \sum_{k=1}^K w_k^{(t)} a_k$$

$$\tilde{x}^{(t)} = \mathcal{F}^{-1}\left(\left(A_x^{lf} + a_{\text{adv}}^{(t)}\right) e^{j\Phi_x}\right)$$

$$\mathcal{L}^{(t)} = \mathcal{L}\left(f(\tilde{x}^{(t)}), y\right)$$

$$u^{(t+1)} = w^{(t)} \odot \exp\left(\eta \nabla_w \mathcal{L}^{(t)}\right)$$

$$w^{(t+1)} = \frac{u^{(t+1)}}{\sum_{j=1}^K u_j^{(t+1)}} \qquad \textit{// simplex normalization}$$

  **end for**

  **return** $\tilde{x}^{(T-1)}$

---

### E. Connections between style and task mismatch.

To directly validate our core premise, we quantify task discrepancy under different corruption types using a frozen source-trained detector with six diagnostic metrics: confidence drop, calibration error, false-positive rate, bounding-box regression error (L1), GIoU error, and Hungarian matching cost. These metrics jointly characterize classification confidence, prediction reliability, localization quality, and assignment difficulty, thereby providing a direct measurement of style-induced task mismatch. As shown in Figure 6, we observe that different corruptions induce distinct discrepancy signatures rather than uniform degradation. Specifically, contrast shift and Gaussian noise mainly amplify classification- and calibration-related discrepancies, motion blur, fog, and pixelation predominantly enlarge localization and matching errors. Snow leads to a more balanced degradation across all task dimensions. Compared with the frozen backbone and static adaptation baseline (Rein), our

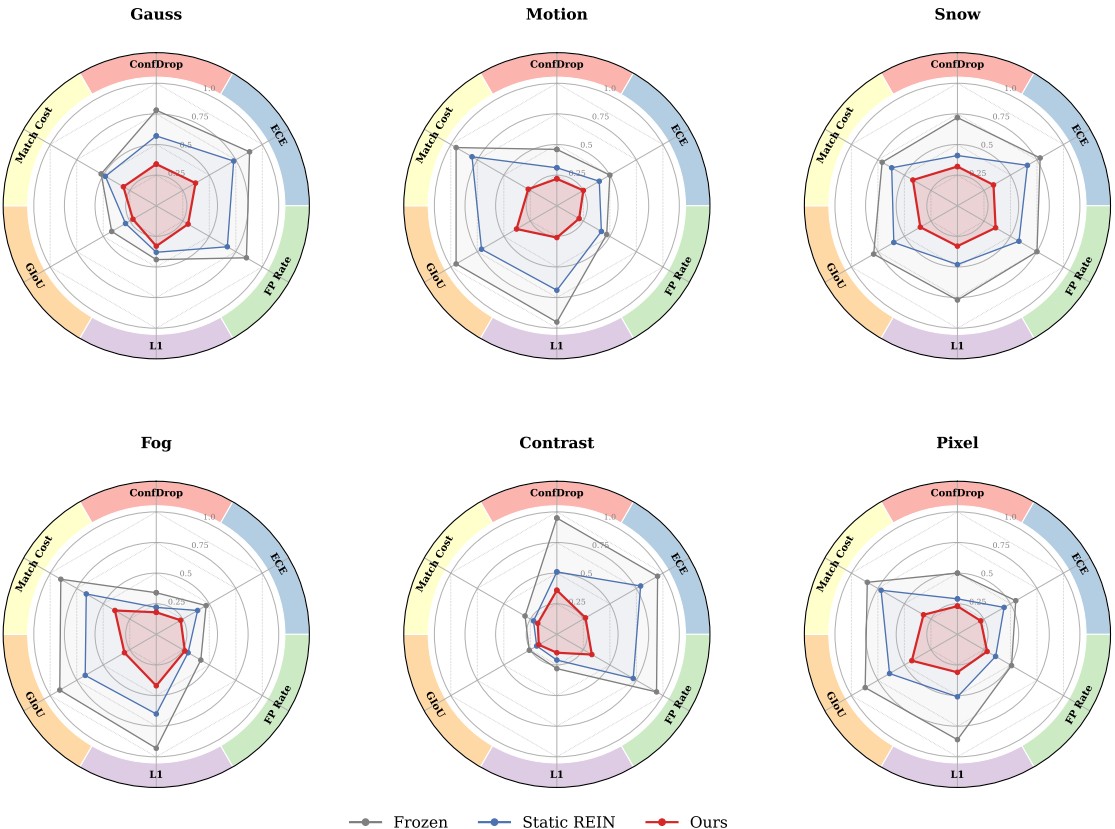

*Figure 6.* Radar plots of style-induced task discrepancy on Cityscapes-C. The six axes correspond to confidence drop(**ConfDrop**), calibration error (**ECE**), false-positive rate (**FP Rate**), bounding-box regression error (**L1**), GIoU error (**GIoU**), and hungarian matching cost (**Matching Cost**). Each axis denotes the relative discrepancy between corruption and source conditions on a specific task dimension. For visualization, each metric is independently normalized to $[0.15, 0.95]$. Smaller values indicate smaller style-induced mismatch.

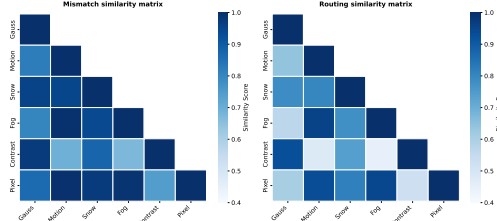

*Figure 7.* Comparison between the mismatch similarity matrix and the routing similarity matrix.

learned routing patterns of our method. The comparison of these two matrices can be visualized in Figure 7. The two matrices exhibit a strong structural alignment: corruption types that are similar in the mismatch space are also grouped similarly in the routing space. Since these corruptions are applied to the same underlying images (i.e., identical semantics with only style perturbations), the observed discrepancy differences can be attributed to style variations rather than semantic changes. This provides direct evidence that the learned style-conditioned routing is not arbitrary, but corresponds to the underlying task discrepancy structure. Similar adaptive interaction and selection mechanisms have also been explored in robust correspondence modeling and cross-modal perception and compositional recognition taskstasks (Cheng et al., 2025b; 2026; Jiang et al., 2025a).

method consistently produces a smaller discrepancy profile and yields stronger reductions on the dominant mismatch dimensions of each style, supporting the effectiveness of our style-aware modeling.

To further examine whether the learned style variable captures task discrepancy, we compute (i) a mismatch similarity matrix based on discrepancy signatures from the frozen detector, and (ii) a routing similarity matrix based on the

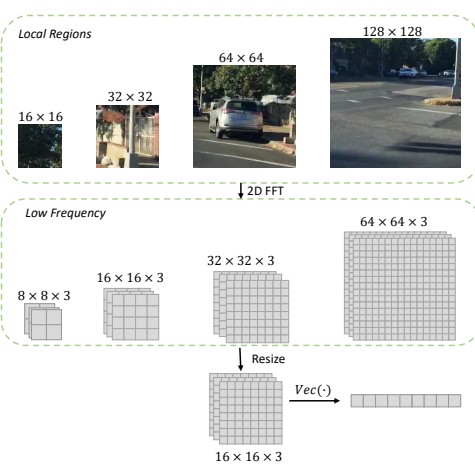

*Figure 8.* Details of spectral pooling.

*Table 11.* Effect of different adaptation designs with varying parameter budgets.

| Method | Params | DF | DR | NR | NC | Avg. |
|---|---|---|---|---|---|---|
| Base | 3.3M | 56.5 | 62.7 | 47.0 | 61.4 | 56.9 |
| Shared Router | 6.1M | 58.4 | 64.3 | 48.5 | 62.9 | 58.5 |
| Random | 8.4M | 56.1 | 62.7 | 46.1 | 60.7 | 56.4 |
| Multiple $A$ + Shared $B$ | 6.0M | 56.9 | 63.1 | 48.2 | 61.9 | 57.5 |
| SCoA (Ours) | 8.4M | **59.3** | **65.1** | **49.8** | **63.5** | **59.4** |

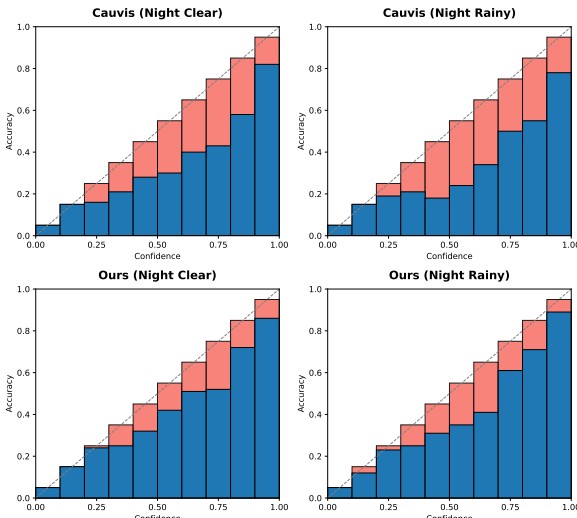

*Figure 9.* Reliability Diagram for different target domains.

# F. Details

## F.1. Adversarial Style Mixing

Inspired by gradient descent with projection onto the simplex (GDPO) (Sagawa et al., 2019), we formulate adversarial style mixing as a simplex-constrained optimization problem as shown in Algorithm 1. Specifically, the mixing weights are restricted to lie on a probability simplex, ensuring a valid convex combination of style memory elements, and are optimized to maximize the detection loss, corresponding to a worst-case style perturbation. We adopt an exponentiated gradient update followed by normalization, which performs gradient descent with implicit projection onto the simplex and preserves the probabilistic interpretation of the mixing weights. In practice, we employ a single optimization step ($T = 1$), which is sufficient to generate strong adversarial style signals while maintaining computational efficiency and training stability.

## F.2. Spectral Pooling

Figure 8 illustrates the spectral pooling operator used to construct the style memory from local image regions. Specifically, we first randomly crop $M = 16$ local regions at four different scales. For each cropped region, we compute the low-frequency amplitude spectrum via 2D FFT followed by spectrum centralization. The extracted low-frequency components are then resized to a fixed resolution of $16 \times 16$ and vectorized to obtain compact style representations for memory construction.

# G. Model Analysis

## G.1. Parameters Analysis

Table 11 compares different adaptation designs under varying parameter budgets. Replacing the style-conditioned router in the full SCoA model with a random counterpart leads to a significant performance drop of 3.0% mAP, despite using the same number of parameters, indicating that the gains of SCoA do not stem from increased model capacity but from effective dynamic adaptation. When removing the layer-wise routing mechanism and using only a shared router across layers, the performance further decreases by 0.9 % mAP, suggesting that layer-wise routing is crucial for capturing layer-specific adaptation patterns. We also explore a variant that employs multiple $A$ matrices with a shared $B$ projection, which results in a noticeable degradation of 1.9% mAP. This observation suggests that learning a shared low-rank basis via the $A$ matrices is more effective.

## G.2. Confidence Analysis

Figure 9 further analyzes the confidence calibration behavior of Cauvis and our method under Night Clear and Night Rainy conditions. Cauvis adopts a causal intervention strategy in the feature space to extract style-invariant representa-

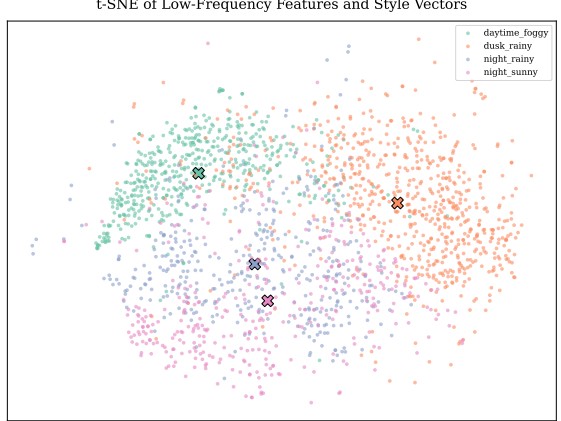

*Figure 10.* t-SNE visualization of low-frequency spectral representations across different domains.

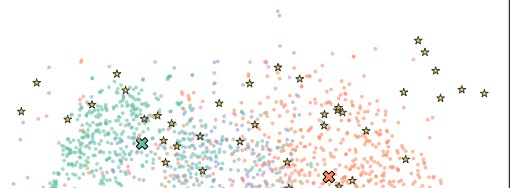

*Figure 11.* t-SNE visualization of style vector distribution.

tions, which effectively suppresses spurious correlations but also inevitably removes style-related discriminative cues. As a consequence, the resulting features tend to be less discriminative, leading to systematically lower confidence scores and under-confident predictions, as reflected by the gap between accuracy and confidence in the reliability diagrams. In contrast, our method does not explicitly discard style-related information, but instead performs style-conditioned task compensation through dynamic adaptation, allowing the model to preserve discriminative capacity.

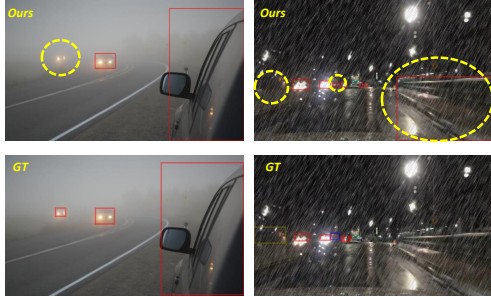

*Figure 12.* Failure Cases. The yellow dashed circles indicate incorrect detection results.

## H. Visualization

### H.1. Diversity of Style Memory

To qualitatively evaluate the diversity of the learned style memory, we randomly sample eight elements from the memory and use them to augment a source-domain image. As shown in Figure 13, the resulting images exhibit noticeably different appearance patterns, including variations in illumination, color tone, contrast, and blur. These diverse transformations indicate that the proposed memory effectively stores heterogeneous style information rather than collapsing to a single dominant mode. Such diversity enables the model to synthesize challenging and varied style priors, which are subsequently leveraged for style-conditioned adaptation in unseen target domains.

Furthermore, we validate the separability of the style representations via t-SNE visualization. Specifically, we randomly sample 500 images from each domain and project their spectral style representations (i.e., low-frequency amplitude statistics) into a 2D space. The visualization is shown in Fig. 10. The results demonstrate that samples from different domains form clearly distinguishable clusters, indicating that spectral amplitude effectively captures

domain-specific appearance variations. Notably, domains with similar global conditions, such as Night Clear and Night Rainy, exhibit partial overlap, which is consistent with their shared illumination characteristics. This suggests that the learned representation reflects meaningful style structure rather than arbitrary separation. We further visualize the learned memory-based style vectors, as shown in Fig. 11. We observe that the learned style vectors tend to cluster around the four evaluation domains, while also extending beyond them. This indicates that the style memory is not restricted to the observed domains, but instead captures a broader style space, with the capacity to represent additional, unseen style variations.

### H.2. Failure Cases

Failure cases are shown in Figure 12. Under severe domain shifts (e.g., dense fog or heavy rain), the degradation of visual cues may still lead to missed detections, particularly for distant or small objects whose boundaries become indistinguishable. Moreover, strong weather-induced noise can occasionally cause false positive detections, where background

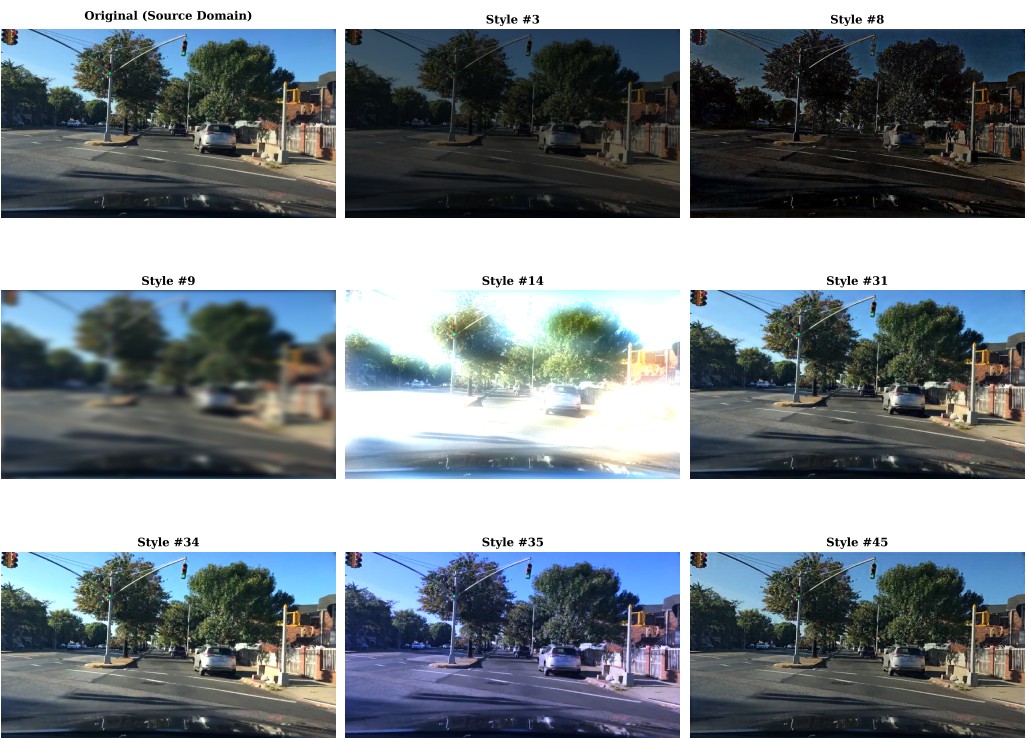

*Figure 13.* Diversity of style memory.

regions are incorrectly identified as foreground objects. These observations suggest that while style-conditioned adaptation improves robustness, it remains challenging to fully handle extreme visibility degradation relying solely on visual information.

### H.3. More Visualization Results

We provide a visualization of the detection results obtained by Rein and our method on five weather conditions in Figure 14-Figure 18. Specifically, as shown in the first column of Figure 14, we observe that our method provides accurate label for the *Truck*, while Rein misclassifies it as *Bus*. This suggests that our method can generate more discriminative features, effectively reducing the false positives caused by misclassification. Furthermore, our approach demonstrates superior capability in distinguishing foreground from background regions, particularly under challenging weather conditions (e.g., the first column in Figure 15).

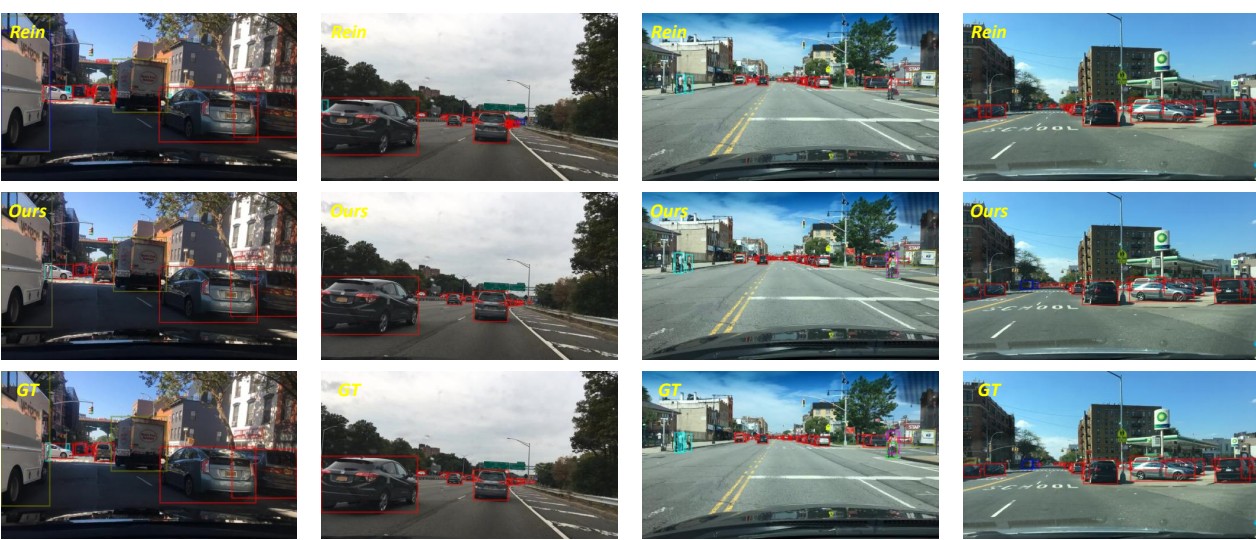

*Figure 14.* Visualization of detection result on Daytime Clear.

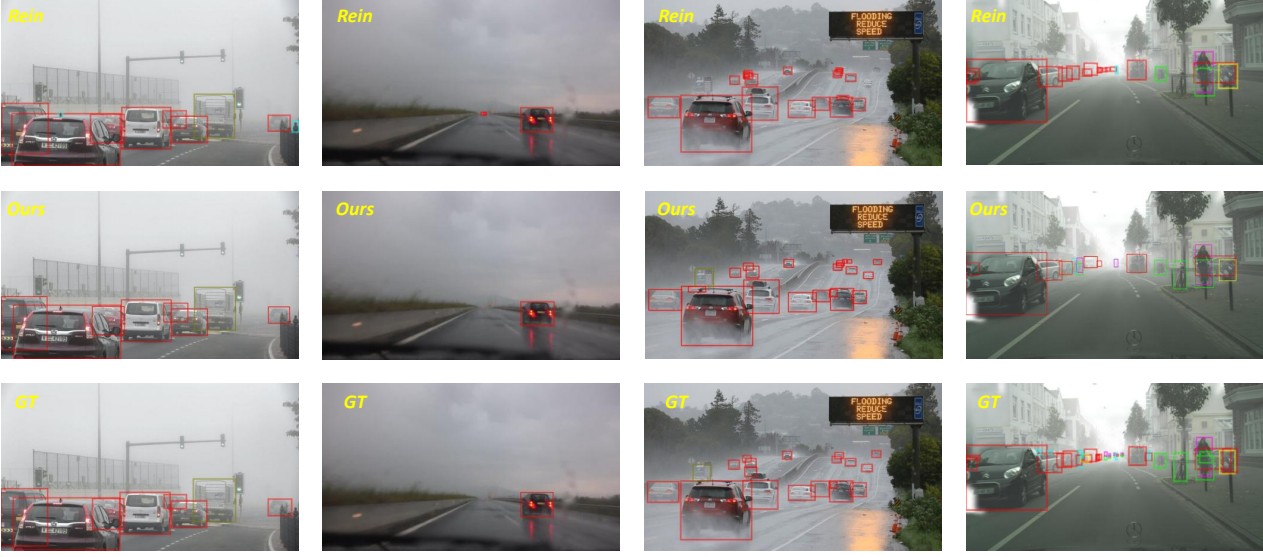

*Figure 15.* Visualization of detection result on Daytime Foggy.

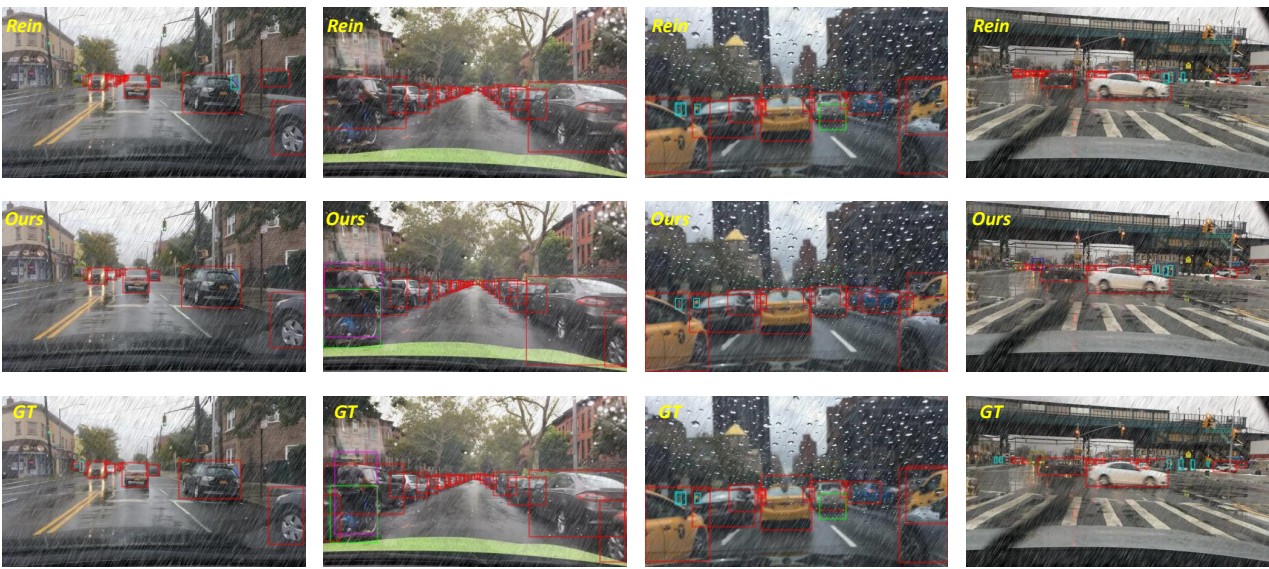

*Figure 16.* Visualization of detection result on Dusk Rainy.

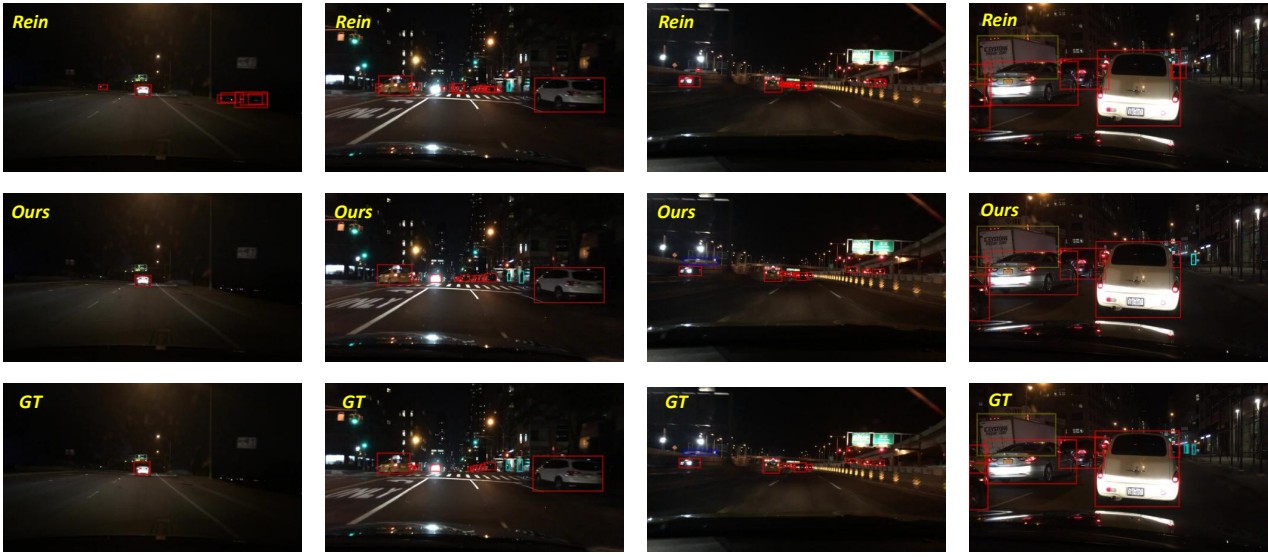

*Figure 17.* Visualization of detection result on Night Clear.

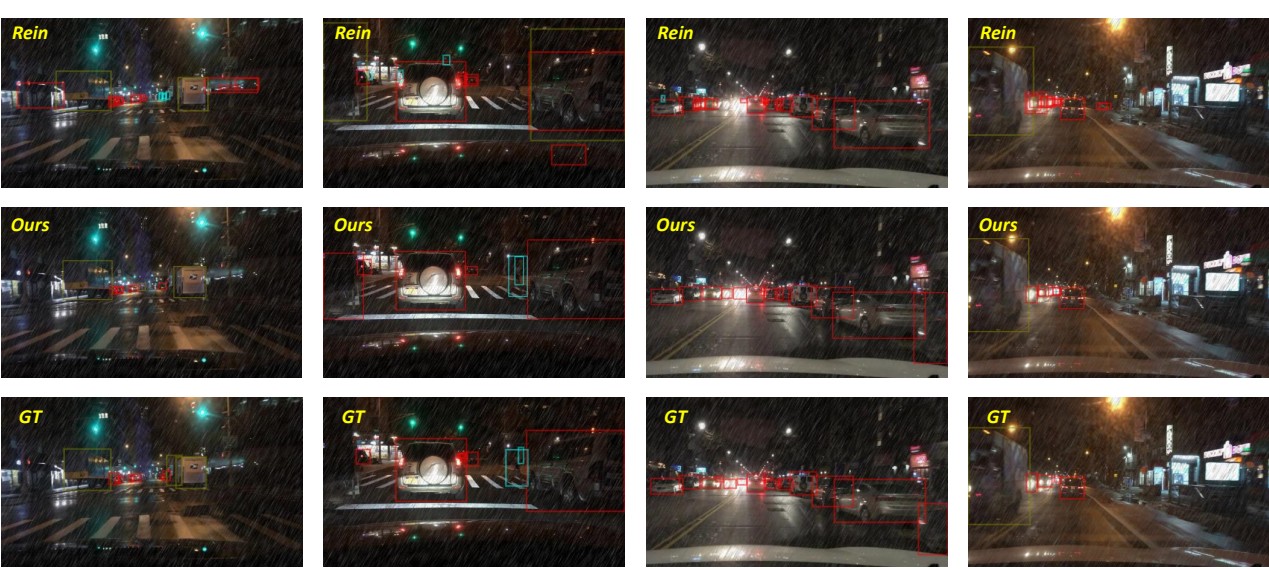

*Figure 18.* Visualization of detection result on Night Rainy.

