# OpenReview forum: "SCoA: Revisiting Domain Generalized Object Detection with Style-Conditioned Adaptation"
_ICML.cc/2026/Conference — ICML 2026 regular_

### Official Review · Reviewer_W5WJ · 2026-03-07

**Soundness:** 3
**Presentation:** 3
**Significance:** 2
**Originality:** 3
**Overall Recommendation:** 4
**Confidence:** 4

**Summary:**

The paper proposes SCoA, a novel style-conditioned adaptation framework for domain-generalized object detection. It addresses the task discrepancy between the pretraining objectives of VFMs and downstream detection tasks caused by diverse visual styles across domains.  The framework consists of three main modules. First, Spectral Style Modeling (SSM) extracts diverse style priors from source domains via Fourier transform, generating a style vector s and a style-augmented image \tilde{x}. Next, in the Mixture-of-Tokens based Adaptation (MTA) module, the style vector s dynamically selects adapter parameters to guide feature adaptation. Finally, Style-Conditioned Query Refinement (SCQR) aligns DETR object queries with the representations produced by MTA. Experiments demonstrate favorable performance across two challenging scenarios.

**Compliance With Llm Reviewing Policy:**

Affirmed.

**Final Justification:**

I will keep the positive score.

**Key Questions For Authors:**

1.How should the hyperparameter l be chosen to sufficiently capture the visual style of the image and the local cropped regions?

2.Please provide qualitative visualizations or quantitative diversity analyses of the learned style memory M_s after training to verify whether the style prototypes \alpha_k well-separated style patterns.

3.Could the authors provide a brief description of the inference procedure in the target domain, including how the style vector $s$ is generated during test time?

4.Could the authors include additional comparisons with methods that have similar or larger numbers of trainable parameters to better isolate the contribution of the proposed method?

**Limitations:**

Yes

**Strengths And Weaknesses:**

Strength:

1.The proposed SCoA presents a clear motivation and well-defined problem formulation, offering a novel perspective on adapting vision foundation models for domain-generalized object detection.

2.The method elegantly integrates multiple techniques—e.g., Fourier-based style modeling, low-rank decomposition, and mixture-of-tokens adaptation—into a coherent and well-structured framework.

Weaknesses:

1.The proposed Spectral Style Modeling (SSM) relies on a low-pass mask M_{lf} to extract low-frequency amplitude components as style representations. However, the cutoff frequency is determined by the hyperparameter l (with d=3l^{2}). The paper lacks a sensitivity analysis on the hyperparameter l.

2.The initialization of the learnable style memory M_s=\{\alpha_k\}^{K}_{k=1} is not clear described in the paper. if M_s is randomly initialized, the \alpha_k may carry no meaningful style semantics, which raise concerns about the reliability of the similarity computation s_{k,m}.

3.The paper does not clearly describe the inference pipeline in the target domain. In particular, it is unclear how the style vector s is generated during test time. If the style vector is directly produced in the target domain without the adversarial optimization, this may lead to a distribution mismatch between the source and target domains.

4.SCoA introduces 8.4M trainable parameters, which is larger than the competing methods (e.g., LoRA and Cauvis). Therefore, it is unclear whether the performance improvement comes from the proposed style-conditioned adap tation mechanism or simply from the increased model capacity.

---

> ### Author Rebuttal · Authors · 2026-03-30
>
> We thank the reviewer for the careful review and constructive feedback.  We will add more baselines for fair comparison in the revised manuscript. Our responses to the main concerns are provided below.
>
> **Q1: Chosen of hyperparameter $l$.**
>
> In SSM, let $a$ denote the side length of each square local region, and  $l$ denote the side length of the retained low-frequency window. By default, we set $l=\frac{1}{2}a$, since low-frequency components are expected to capture the dominant style information. We further evaluate other choices, including$l=\frac{1}{8}a$, $l=\frac{1}{4}a$, $l=\frac{3}{4}a$, and $l=a$.  As shown in the table below,  a smaller $l$ cannot preserve enough style cues, while an excessively large $l$ tends to introduce more content-related details.
>
> | $l$ | $\frac{1}{8}a$ | $\frac{1}{4}a$ | $\frac{1}{2}a$ | $\frac{3}{4}a$ | $a$ |
> |-----|---------------|---------------|---------------|---------------|-----|
> | mAP | 58.7 | 59.2 | **59.4** | 59.1 | 58.4 |
>
> **Q2:  Qualitative visualizations of style memory.**
>
>  The style memory $M_s$ is initialized with Kaiming initialization to ensure a well-dispersed starting point.   The t-SNE visualizations of style memory can be  found in anonymous GitHub repository: https://anonymous.4open.science/r/rebuttal-7954/Figure5.png, which shows that the learned prototypes form well-separated clusters. This indicates that $M_s$ captures diverse and discriminative style patterns rather than collapsing to trivial solutions.
>
> **Q3:  Inference procedure.**
>
> During inference, the style vector $s$ is directly extracted from the input image **without any adversarial optimization**. Specifically, we compute the low-frequency amplitude component of the test image and feed it into the style encoder $h(\cdot)$ to obtain $s$, which is then used by the router for dynamic adaptation. The whole inference pipeline is available at this link: https://anonymous.4open.science/r/rebuttal-7954/Alg1.png.
>
> We would like to emphasize that the adversarial style mixing can be viewed as a form of distributionally robust optimization, exposing the model to challenging style perturbations within the source domain. At test time, the style vector is derived from the actual input image, which naturally lies within or is better covered by the expanded style space learned during training.
>
> **Q4:  Comparison with other methods.**
>
> First, we compare SCoA with several parameter-efficient fine-tuning methods, including LoRA and PiSSA. As shown in the table below, even with a larger parameter budget (e.g., LoRA (r=32) and PiSSA with 10.1M parameters), these methods still underperform SCoA (8.4M). indicating that the gain cannot be explained by increased model capacity alone.
>
> Second, a substantial portion of SCoA’s parameters arises from the layer-wise routers and the cross-attention modules in SCQR, which are specifically designed for style-conditioned adaptation. To isolate the effect of capacity, we further introduce **a lightweight variant, SCoA (light)**, by removing SCQR and adopting a shared router. Despite using fewer parameters (4.3M), SCoA (light) still outperforms all competing methods, including Cauvis (4.7M) , demonstrating that the performance gain stems from the proposed style-conditional adaptation mechanism rather than parameter scaling.
>
>
> | Method           | Params | DF   | DR   | NR   | NC   | Avg.  |
> |------------------|--------|------|------|------|------|-------|
> | LoRA (r=16)      | 5.5    | 49.5 | 58.1 | 46.1 | 59.6 | 53.3  |
> | LoRA (r=32)      | 10.1   | 49.8 | 59.0 | 45.4 | 61.3 | 53.9  |
> | PiSSA [1]    | 10.1   | 52.3 | 61.3 | 46.8 | 61.9 | 55.6  |
> | Cauvis           | 4.7    | 56.5 | 64.6 | 47.6 | 61.2 | 57.5  |
> | SCoA (light)     | 4.3    | 58.2 | 64.3 | 48.2 | 62.4 | 58.3  |
> | **SCoA**         | **8.4**| **59.3** | **65.1** | **49.8** | **63.5** | **59.4** |
>
> [1] Fanxu Meng, Zhaohui Wang, and Muhan Zhang. Pissa: Principal singular values and singular vectors adaptation of large
> language models. In NeurIPS, 2024.

---

> > ### Author Rebuttal · Reviewer_W5WJ · 2026-04-01
> >
> > The author essentially answered my questions, but due to the large number of parameters in the algorithm, I opted for a partial response.

---

> > > ### Author Response · Authors · 2026-04-03
> > >
> > > We sincerely thank the reviewer for the follow-up comment. Due to time constraints during the initial rebuttal phase, comparisons with additional larger-parameter PEFT methods were not included. We have now added these experiments, and the results are summarized below.
> > >
> > > | Method          | Params   | DF       | DR       | NR       | NC       | Avg.     |
> > > | --------------- | -------- | -------- | -------- | -------- | -------- | -------- |
> > > | ER-LoRA[1]        | 8.7M     | 52.3     | 60.9     | 47.4     | 61.3     | 55.5    |
> > > | DA-VPT[2]          | 13.6M    | 58.8     | 63.1     | 48.3     | 62.1     | 58.1    |
> > > | SPT-Adapter[3]     | 14.6M    | 57.8     | 63.1     | 46.3     | 61.9     | 57.3    |
> > > | SPT-LoRA [3]       | 14.6M    | 56.9     | 63.2     | 47.0     | 62.1     | 57.3    |
> > > | SCoA-random      | 8.4M    | 57.7     | 63.2     | 45.7     | 56.5     | 55.8    |
> > > | **SCoA (ours)** | **8.4M** | **59.3** | **65.1** | **49.8** | **63.5** | **59.4** |
> > >
> > > As shown in the first four rows, methods with substantially larger parameter budgets (8.7M–14.6M) still underperform SCoA (8.4M). This suggests that simply increasing model capacity is insufficient to explain the observed performance gains.
> > >
> > > From a methodological perspective, these approaches mainly improve **static adaptation mechanisms**:
> > > ER-LoRA performs weight-level low-rank adaptation with adaptive rank selection; DA-VPT introduces prompt-based semantic alignment; and SPT-based methods allocate adaptation strength based on layer-wise importance. However, all these methods learn **a shared adaptation function** across inputs, without explicitly modeling style-dependent variations.
> > >
> > > In contrast, our method is built upon a different principle: we explicitly model style-dependent task discrepancy and perform style-conditioned adaptation through dynamic routing. This allows the model to adapt its behavior according to the input style, instead of relying on a single global adaptation strategy.
> > >
> > > To further verify that the gain comes from correct style conditioning rather than the architecture itself, we conduct a controlled experiment where the model is trained normally, but at inference time the style vector is randomly sampled from the style memory for each input (termed as **SCoA-random**). This preserves the learned parameters and model capacity, while breaking the correspondence between the input and its style representation. We observe a clear performance drop compared to the full model, indicating that correct style conditioning is essential for achieving strong performance.
> > >
> > > Overall, these results indicate that the advantage of SCoA arises from its ability to perform dynamic, style-aware adaptation, rather than from increased parameter count or stronger static designs.
> > >
> > > We hope this addresses the reviewer’s concern and clarifies our contribution. We will include these results in the final version.
> > >
> > >
> > > [1] ER-LoRA: Effective-Rank Guided Adaptation for Weather-Generalized Depth Estimation. arXiv:2509.00665, 2025.
> > >
> > > [2] DA-VPT: Semantic-guided visual prompt tuning for vision transformers. CVPR, 2025.
> > >
> > > [3] Sensitivity-aware visual parameter-efficient fine-tuning. ICCV,2023.

---

### Official Review · Reviewer_LJMZ · 2026-03-12

**Soundness:** 4
**Presentation:** 4
**Significance:** 2
**Originality:** 3
**Overall Recommendation:** 4
**Confidence:** 4

**Summary:**

This paper introduces SCoA, a framework designed for Domain Generalized Object Detection (DGOD) by leveraging the representational power of Vision Foundation Models (VFMs). The proposed SCoA framework integrates three primary components:
- Spectral Style Modeling (SSM): This module characterizes visual styles in the frequency domain using Fast Fourier Transform (FFT). It maintains a learnable style memory to store low-frequency amplitude components from local image regions. By employing an adversarial weighting mechanism, the module synthesizes challenging and diverse style signals from a single source domain to serve as conditioning priors.
- Mixture-of-Tokens Adaptation (MTA): To achieve flexible adaptation, the authors design a dynamic token-based mechanism. Adaptation tokens are parameterized via a low-rank decomposition, comprising a shared matrix for common task knowledge and multiple style-specific matrices. A lightweight router, conditioned on the extracted style vector, dynamically selects and composes these matrices for each input.
- Style-Conditioned Query Refinement (SCQR): Recognizing that detection queries in DETR-like heads should also be style-aware, this module aggregates tokens across multiple backbone layers. It uses cross-attention to inject style-specific information into the object queries, transforming them from static entities into adaptive counterparts that better handle style-induced task mismatches.

**Compliance With Llm Reviewing Policy:**

Affirmed.

**Final Justification:**

The authors have addressed my concerns in the rebuttal and have thoroughly discussed the limitations. Therefore, I am willing to give a final score of 4.

**Key Questions For Authors:**

1. Given that DINOv2 is itself pre-trained on large-scale, multi-source data, it is unclear whether this performance gap is attributable to the absence of style-conditioned adaptation specifically, or simply to the sub-optimal practice of fine-tuning the detection head on a single-source domain. A more informative and fair baseline would be to fine-tune the DINOv2 detection head with strong input-level style augmentation strategies, such as FDA[1].
2. While the paper reports parameter counts, it provides no analysis of the resulting inference latency or FLOPS overhead.
3. The proposed SSM relies on a style memory constructed exclusively from local regions of the single source domain, and generates novel styles via adversarial mixing of these stored entries. This raises a fundamental concern about generalization: if the target domain exhibits appearance characteristics that are markedly different from those of the source domain (e.g., remote-sensing imagery or underwater scenes), the style memory may fail to produce meaningful or relevant conditioning signals. In such cases, would the style-conditioned router actively mislead the adaptation mechanism?
4. How about the performance on objects of different sizes?

I am happy to raise my rating if questions 1 and 3 are solved.

[1] FDA: Fourier Domain Adaptation for Semantic Segmentation, CVPR 2020

**Limitations:**

yes

**Strengths And Weaknesses:**

- Soundness
The paper is generally technically sound, with well-designed ablation studies and consistent results across multiple backbones.
- Presentation
The paper is clearly written and well-organized.
- Significance
The paper addresses a timely and practically relevant problem in autonomous driving and robotics. However, as foundation models continue to scale and internalize increasingly robust representations, the necessity of per-sample style-conditioned adaptation may become less compelling over time.
- Originality
SCoA does not introduce fundamentally new operators, but offers a well-motivated combination of spectral style synthesis, MoE-based adaptation, and style-conditioned query refinement for DGOD.

---

> ### Author Rebuttal · Authors · 2026-03-30
>
> We sincerely thank the reviewer for the valuable comments. In the revised version, we will further clarify the behavior of our model under extreme domain shifts. Our detailed responses to the main concerns are provided below.
>
> **Q1: Comparison with strong augmentation strategy.**
>
> Following your suggestion, we apply FDA-based style augmentation to DINOv2. As shown in the table below, FDA yields only a marginal improvement of $0.3$ over the baseline, indicating limited benefit from input-level style augmentation.
>
> We further combine FDA with a static PEFT method (Rein). Both FDA and our SSM bring only minor improvements in this setting, suggesting that simply increasing style diversity is insufficient for well-pretrained VFMs. In contrast, SCoA achieves substantially stronger performance, indicating that the gain primarily comes from style-conditioned feature adaptation rather than input-level augmentation. We will include these baselines in the revised paper.
>
>
> | Method                                   | DF   | DR   | NR   | NC   | Avg.  |
> |------------------------------------------|------|------|------|------|-------|
> | DINOv2                                  | 53.5 | 60.8 | 42.6 | 59.5 | 54.1  |
> | DINOv2 + FDA                            | 54.1 | 61.0 | 43.4 | 59.2 | 54.4  |
> | DINOv2 w/ token-based adapter           | 54.7 | 62.6 | 44.9 | 59.1 | 55.3  |
> | DINOv2 w/ token-based adapter + SSM     | 55.3 | 62.6 | 45.2 | 59.6 | 55.7  |
> | DINOv2 w/ token-based adapter + FDA     | 55.1 | 62.8 | 45.2 | 59.2 | 55.6  |
> | **SCoA**                                | **59.3** | **65.1** | **49.8** | **63.5** | **59.4** |
>
> **Q2: Computational overhead.**
>
> Please refer to our response to Reviewer Rcf6 (Q3) due to space limitations.
>
> **Q3: Generalization on extremely different style datasets.**
>
> We thank the reviewer for this insightful concern. We address it from two aspects.
>
> **(1) Robustness under severe style shifts.**
> Although no underwater benchmark with the same object categories is available, we construct three synthetic underwater variants with different severity levels to simulate strong appearance shifts.  Visual examples are provided in anonymous GitHub repository: https://anonymous.4open.science/r/rebuttal-7954/Figure8.png. Compared with daytime scenes, underwater images exhibit low illumination, reduced contrast and blur. As shown in the table below, SCoA outperforms the baseline across all severity levels, with larger gains under more severe degradations, indicating the effectiveness of style-conditioned adaptation in underwater-style shifts.
>
> For remote-sensing imagery, the domain gap is mainly driven by layout and viewpoint variations rather than style changes, making it less aligned with the focus of our method.
>
> | Method   | Mild | Medium | Strong |
> |----------|------|--------|--------|
> | baseline | 51.3 | 44.3   | 32.1   |
> | SCoA     | 57.5 | 52.2   | 45.6   |
>
> **(2) What happens when the style memory does not cover a target style?**
>  To study this, we visualize the low-frequency spectral statistics of four target domains (DF, DR, NR, NC) at this link: https://anonymous.4open.science/r/rebuttal-7954/Figure4.png,  and observe that DF is clearly separated from the other three domains. Based on this, we construct an experiment where the style memory is replaced with low-frequency statistics sampled only from DR/NR/NC,  simulating a case where the memory fails to cover DF-style patterns (denoted as **Model A**). As shown in the table below, this leads to a noticeable performance drop on DF.
> | Method  | mAP  |
> |-----|------|
> | Frozen  | 53.5 |
> | Model A | 55.9 |
> | SCoA    | 59.3 |
>
> To further analyze this phenomenon, we visualize the routing weights (https://anonymous.4open.science/r/rebuttal-7954/Figure6.png) and observe that they become significantly smoother. This indicates that the router loses discriminative, style-specific behavior and degenerates into a static PEFT mechanism.  We further analyze task mismatch using radar plots (https://anonymous.4open.science/r/rebuttal-7954/Figure7.png). In the foggy domain, the mismatch is mainly reflected in degraded localization accuracy due to blur. Although overall performance still improves, domain-specific discrepancies remain largely unaddressed.
>
> **Q4: Performance on different sizes.**
>
> We evaluate the performance across different object scales as shown in the table below, and our method improves detection performance across all sizes.
>
> | Method   | DF-s | DF-m | DF-l | DF | DR-s | DR-m | DR-l | DR | NR-s | NR-m | NR-l | NR| NC-s | NC-m | NC-l | NC |
> |----------|------|------|------|--------|------|------|------|--------|------|------|------|--------|------|------|------|--------|
> | baseline | 10.5 | 44.9 | 74.3 | 54.7   | 17.9 | 52.3 | 77.7 | 62.6   | 10.2 | 39.6 | 65.5 | 44.9   | 16.0 | 57.8 | 76.6 | 59.1   |
> | Ours     | 14.7 | 46.8 | 78.9 | 59.3   | 20.4 | 53.9 | 82.5 | 65.1   | 15.4 | 44.3 | 68.1 | 49.8   | 20.1 | 62.9 | 78.3 | 63.5   |

---

> > ### Author Rebuttal · Reviewer_LJMZ · 2026-04-01
> >
> > Thank you for your response. My concerns have been fully addressed, so I have decided to raise my rating from 3 to 4. I suggest discussing the following point in the revision: what happens when the style memory does not cover a target style.

---

> > > ### Author Response · Authors · 2026-04-02
> > >
> > > We sincerely thank you for your thorough review and constructive comments.
> > >
> > > As suggested, we will incorporate the discussion in the final version to address cases where the style memory does not adequately cover target styles.
> > > Based on your valuable feedback, we believe that the revised manuscript will further improve in overall quality.
> > >
> > > We thank you again for your insightful review and guidance.

---

### Official Review · Reviewer_Rcf6 · 2026-03-12

**Soundness:** 3
**Presentation:** 3
**Significance:** 2
**Originality:** 2
**Overall Recommendation:** 4
**Confidence:** 3

**Summary:**

This paper studies domain generalization for object detection under domain shift. The authors propose a framework that extracts domain style representations from the spectral amplitude statistics of input images and uses them to guide feature adaptation in the detection model. Specifically, the method first transforms images into the frequency domain and derives a style vector from the amplitude spectrum to capture domain-specific appearance characteristics. This style vector is then used to modulate intermediate feature representations through a style-guided adaptation module. The framework further includes additional adaptation and refinement components to improve robustness to domain shift. Experiments are conducted on cross-domain detection benchmarks to evaluate the effectiveness of the proposed approach.

**Compliance With Llm Reviewing Policy:**

Affirmed.

**Final Justification:**

The rebutall has addressed my concerns, I keep my prior positive assessment.

**Key Questions For Authors:**

Q1. Why is spectral amplitude chosen as the representation of style?
The paper assumes that spectral amplitude statistics encode domain-specific style information. Could the authors provide more intuition or empirical evidence supporting this choice? For example: Are there visualizations of the learned style vectors? Do different domains produce clearly separable spectral statistics?

Q2. Clarification of the AdaIN baseline in Table 5. The paper reports comparisons with AdaIN, but it is unclear whether the AdaIN baseline is evaluated under the same pipeline as the proposed method. Specifically, does the AdaIN baseline include the same MOT adaptation and refinement modules? Or does this table only evaluate the effect of different augmentations, without using subsequent operations?

Q3. What is the computational overhead of the proposed modules? Since the method introduces spectral transformations and style extraction modules, it would be helpful to report runtime or FLOPs comparisons with baseline methods.

**Limitations:**

yes

**Strengths And Weaknesses:**

Strengths
1. A unified framework combining spectral style extraction and feature adaptation. The proposed pipeline integrates spectral style extraction, style-guided feature modulation, and refinement modules into a single framework.

2. Empirical results demonstrate improvements over several baselines.
The experiments show improvements over some existing approaches (e.g., VLLM+CLIP+Adap), suggesting that the proposed mechanism may capture domain-specific appearance statistics useful for detection.

Weakness
1. The connection between spectral style and feature adaptation is insufficiently justified. The method extracts style vectors from spectral amplitude statistics and uses them to guide feature adaptation. However, the paper does not clearly explain why spectral amplitude statistics provide a meaningful representation of domain shift for object detection. There remains a semantic gap between pixel-level spectral style and high-level feature adaptation, and the paper provides limited theoretical or empirical evidence to bridge this gap.

2. Limited experimental analysis of the spectral style representation. The effectiveness of the spectral style representation is mainly validated through performance improvements. However, the paper lacks deeper analysis of the learned style vectors. For example:
- No visualization of the spectral style vectors distribution.
- No analysis showing that spectral amplitude statistics correlate with domain shift.

3. Computational overhead is not analyzed.
The proposed method introduces additional modules including spectral transformation, style extraction, and feature modulation. However, the paper does not provide analysis of:
- computational cost
- runtime overhead
- memory consumption

compared to simpler adaptation mechanisms such as AdaIN or feature-level normalization, the efficiency trade-offs remain unclear.

---

> ### Author Rebuttal · Authors · 2026-03-30
>
> We thank the reviewer for the valuable comments on our manuscript. We will make a deeper analysis of style vectors in the revised version.  Our responses to the main concerns are provided below.
>
>
> **W1: Connections between spectral style and feature adaptation.**
>
> We clarify that the spectral style representation is  **not used to directly perform feature adaptation.**  Instead, the extracted style vectors serve as **a conditioning signal** for the router, which dynamically selects and combines learnable adaptation tokens in the feature space. The actual adaptation is performed by these tokens, while the spectral style acts as an indexing signal for selecting appropriate, style-specific adaptation behaviors.
>
> To empirically validate this design, we compare two pairwise similarity structures:
> (1) similarity of spectral style vectors, and (2) similarity of routing distributions learned by our model.
> Concretely, we randomly sample 5 images from each evaluation domain and compute pairwise similarities across all  sampled images in both spaces, respectively. The resulting matrices (see anonymous Github repository: https://anonymous.4open.science/r/rebuttal-7954/Figure3.png) exhibit closely aligned structures: images that are similar in  spectral-style space are also routed to similar adaptation patterns.
> This shows that spectral amplitude statistics are not arbitrary low-level descriptors, but capture domain variations that are directly relevant to downstream detection behavior.
>
> **W2&Q1: Visualizations of style representation.**
>
> **Spectral style distribution.** We first randomly sample 500 images from each domain and visualize their spectral style representations (i.e., low-frequency amplitude statistics) using t-SNE. The t-SNE visualization can be found at: https://anonymous.4open.science/r/rebuttal-7954/Figure4.png.
> The results show that samples from different domains form clearly distinguishable clusters, indicating that spectral amplitude effectively captures domain-specific appearance variations. Notably, domains with similar global conditions, such as **night clear** and **night rainy**, exhibit partial overlap, which is consistent with their shared illumination characteristics. This suggests that the representation reflects meaningful style structure rather than arbitrary separation.
>
> **Learned style vectors.** We further visualize the learned memory-based style vectors. The visualization can be available at https://anonymous.4open.science/r/rebuttal-7954/Figure5.png. We observe that the learned style vectors cluster around the four evaluation domains, while also extending beyond them. This indicates that the style memory is not limited to these domains, but captures a broader style space, with the capacity to represent additional, unseen style variations.
>
> In addition, *Figure 7 in Appendix G.1* shows that different memory elements induce diverse appearance changes (illumination, color tone, contrast, and blur), suggesting that low-frequency spectral statistics capture fundamental style attributes. Moreover, these attributes can be composed to model more complex domain shifts, such as foggy conditions, which involve joint changes in illumination, contrast, and visibility.
>
> **Overall**, the above results demonstrate that:
> (1) low-frequency spectral representations effectively characterize domain variations; and (2) the learned style vectors cover a broad spectrum of potential test domains.
>
>
> **Q2: Clarification of  Table 5.**
>
> In Table 5 of the main paper, the AdaIN baseline follows the same pipeline as our method, with the only difference that the style in SSM is extracted using AdaIN. Specifically, we represent the style by concatenating the channel-wise mean and variance of the image features. All subsequent modules (e.g., MTA and SCQR) remain unchanged, ensuring a fair comparison.
>
>
> **W3&Q3: Computational overhead.**
>
> We provide a quantitative comparison of FLOPs, FPS, GPU memory, and training time in the table below. Compared to other methods, our SCoA introduces additional computational overhead, but also brings consistent performance improvements.
>
> | Method                              | FLOPs | FPS | GPU Memory (G) | Training Time (hrs) | mAP  |
> |-------------------------------------|-------|-----|------------|------------|------|
> | Frozen Dino                         | 228.4 | 4.2 | 9.8       |9.5| 54.1 |
> | Frozen Dino + feature normalization | 228.6 | 4.3 |9.8|  9.5  | 54.5 |
> | Rein                                | 229.7 | 4.6 | 10.0      | 9.9| 55.5 |
> | Cauvis                              | 282.5 | 4.7 | 11.2       | 14.2| 57.5 |
> | AdaIN                               | 238.5 | 5.2 |  12.0 |11.8| 59.0 |
> | **Ours**                            | 238.8 | 5.3 | 12.0 |11.8|  59.4 |

---

> > ### Author Rebuttal · Reviewer_Rcf6 · 2026-04-04
> >
> > My concerns have been adequately addressed, so I would like to keep my positive score.

---

> > > ### Author Response · Authors · 2026-04-04
> > >
> > > We sincerely thank you  for the valuable feedback and positive support.
> > >
> > > Following your suggestions, we will include additional visualizations and a more comprehensive analysis of computational overhead in the revised version. We believe these revisions will further enhance the overall quality of the manuscript.
> > >
> > > Thank you again for your thoughtful review and valuable guidance.

---

### Official Review · Reviewer_b8Y3 · 2026-03-13

**Soundness:** 3
**Presentation:** 2
**Significance:** 3
**Originality:** 2
**Overall Recommendation:** 4
**Confidence:** 3

**Summary:**

This paper proposes SCoA, a style-conditioned adaptation framework for domain generalized object detection built on frozen vision foundation models. The method combines three components: a spectral style modeling module that builds a frequency-domain style memory and adversarially mixes styles, a mixture-of-tokens adapter routed by a style vector, and a style-conditioned query refinement module that injects style information into object queries. Empirically, the paper reports gains over prior PEFT-style baselines on Urban Scene and on Cityscapes-C corruptions, with the strongest main-table result improving average Urban Scene mAP from 57.5 to 59.4 over Cauvis on DINOv2-L, and mPC on Cityscapes-C from 35.6 to 40.4.

**Compliance With Llm Reviewing Policy:**

Affirmed.

**Final Justification:**

My concerns have been adequately addressed, so I would like to raise my rating from 3 to 4. I would like to see the discussion in the revision: Connections between style and task mismatch.

**Key Questions For Authors:**

No

**Limitations:**

The work is limited by narrow benchmark coverage in the main paper, weak evidence for the central conceptual claim, and substantial ambiguity about fairness of some comparisons. The method also appears fairly complex relative to the observed incremental gains of several components.

**Strengths And Weaknesses:**

**Strengths**

1. The paper targets a meaningful and practical DGOD setting: training on a single source domain and generalizing to unseen domains using frozen VFMs plus PEFT-style adaptation. The motivation is timely and relevant for robust detection.
2. The empirical results are solid overall. On Urban Scene with DINOv2-L, SCoA improves average mAP from 57.5 to 59.4 over Cauvis; across EVA02, SAM, and DINOv2 backbones, it also improves average mAP relative to Cauvis in Table 2.
3. The ablation section is better than average. The paper isolates SSM, MTA, and SCQR; it also compares simplified token/attention modulation variants and shuffled routing, which supports the claim that routing matters.

**Weaknesses**

Originality is combination. The paper mainly combines existing motifs: FFT-based style statistics, memory-based style augmentation, adversarial mixing, routed expert/token adaptation, low-rank parameterization, and query refinement via cross-attention. The claimed conceptual shift from “static discrepancy” to “dynamic style-dependent discrepancy” is more a framing device than a clearly new technical formulation.

The paper does not convincingly validate its core premise. The central claim is that different styles induce different task mismatches, but the evidence is mostly indirect: routing histograms and performance gains. There is no direct measurement showing that the learned style variable corresponds to the claimed task discrepancy, nor that style is the dominant factor behind the gain.

---

> ### Author Rebuttal · Authors · 2026-03-30
>
> We thank the reviewer for the valuable comments and suggestions on our manuscript. We will further clarify our motivation for task mismatch in the revised version.  Our responses to the concerns are provided below.
>
>  **W1: About Originality.**
>
> We would like to clarify that our contribution does not lie in introducing a single new module, but in **formulating and addressing a previously underexplored problem in DGOD.**
>
> **(1) Novel formulation.**
> Existing VFM-based methods assume a static task discrepancy between pretraining and detection, leading to style-agnostic adaptation strategies. In contrast, we show that task discrepancy is inherently style-dependent (e.g., fog vs. Gaussian noise affecting localization vs. classification differently), motivating dynamic, style-conditioned task compensation, which has not been explicitly studied in prior DGOD work.
>
> **(2) Non-trivial combination.**
> Our framework is not a direct combination of existing techniques, but a **co-designed pipeline**  aligned with the above formulation:
> - SSM provides a style indexing signal that captures appearance variations relevant to task mismatch.
> - MTA performs conditional adaptation via routing, enabling different samples to follow different adaptation trajectories.
> - SCQR injects style information into detection queries, addressing the limitation of static detection heads.
>
> These components are tightly coupled through style conditioning rather than independently applied.
>
> **(3) Style-aware parameterization.**
> MTA adopts a MoE token design with low-rank decomposition,   separating adaptation into a shared domain-invariant component and multiple style-specific components. While prior PEFT methods introduce adapters or prompts, they typically learn a single unified adaptation function under the source domain, without explicitly modeling style-dependent variation.
>
> **(4) Dynamic adaptation rather than module stacking.**
> We design a lightweight variant using only predefined style augmentation[1] and a shared router in MTA. As shown in the table below, despite its simplicity, this variant still achieves strong performance, indicating that the gain arises from style-conditioned adaptation rather than module stacking.
>
> | Method        | Params | DF   | DR   | NR   | NC   | Avg. |
> |--------------|--------|------|------|------|------|------|
> | Rein         | 3.0      | 55 .0  | 62.4 | 45.2 | 59.4 | 55.5 |
> | SoMA         | 4.9    | 51.0   | 59.3 | 47.6 | 59.3 | 54.3 |
> | SCoA (light) | 4.1    | 57.1 | 64.1 | 48.1 | 61.8   | 57.8 |
>
> **W2: Connections between style and task mismatch.**
>
> To directly validate our core premise, we quantify task discrepancy under different corruption types using a frozen source-trained detector with six diagnostic metrics: confidence drop, calibration error, false-positive rate, bounding-box regression error (L1), GIoU error, and Hungarian matching cost. These metrics jointly characterize classification confidence, prediction reliability, localization quality, and assignment difficulty, thereby providing a direct measurement of style-induced task mismatch. The corresponding figure can be found at https://anonymous.4open.science/r/rebuttal-7954/Figure1.png.
> We observe that  different corruptions induce **distinct discrepancy signatures rather than uniform degradation.**
> Specifically, contrast shift and Gaussian noise mainly amplify classification- and calibration-related discrepancies, motion blur, fog, and pixelation predominantly enlarge localization and matching errors. Snow leads to a more balanced degradation across all task dimensions. Compared with the frozen backbone and static adaptation baseline (Rein), our method consistently produces a smaller discrepancy profile and yields stronger reductions on the dominant mismatch dimensions of each style, supporting the effectiveness of our style-aware modeling.
>
> To further examine whether the learned style variable captures task discrepancy, we compute (i) a **mismatch similarity matrix** based on discrepancy signatures from the frozen detector, and (ii) a **routing similarity matrix** based on the learned routing patterns of our method. The comparison of these two matrices can be visualized at this link: https://anonymous.4open.science/r/rebuttal-7954/Figure2.png.
> The two matrices exhibit a strong structural alignment: corruption types that are similar in the mismatch space are also grouped similarly in the routing space.
> Since these corruptions are applied to the same underlying images (i.e., **identical semantics with only style perturbations**), the observed discrepancy differences can be attributed to style variations rather than semantic changes. This provides direct evidence that the learned style-conditioned routing is **not arbitrary**, but corresponds to the underlying task discrepancy structure.
>
> [1] Physaug: A physical-guided and frequency-based data augmentation for single-domain generalized object detection.  AAAI, 2025.

---

> > ### Author Rebuttal · Reviewer_b8Y3 · 2026-04-07
> >
> > My concerns have been adequately addressed, so I would like to raise my rating from 3 to 4. I would like to see the discussion in the revision: Connections between style and task mismatch.

---

> > > ### Author Response · Authors · 2026-04-07
> > >
> > > We sincerely thank the reviewer for the insightful comments.
> > >
> > > In the revised version, we will further elaborate on the relationship between style and task mismatch. This will help us more convincingly validate our core premise.
> > >
> > > We again thank the reviewer for the time and effort devoted to reviewing our work.

---

### Decision · Program_Chairs · 2026-04-30

**Decision:**

Accept (regular)

**Comment:**

The paper proposes SCoA. This style-conditioned adaptation mechanism for generalized object detection builds on foundation models.
As the authors point out, recent advances in domain-generalized object detection (DGOD) have increasingly exploited vision foundation models (VFMs) via parameter-efficient finetuning, but that they adapt VFMs with fixed, style-agnostic parameters, overlooking that different visual styles may induce distinct task discrepancies. Conditioned on style signals, a Mixture-of-Tokens Adaptation (MTA) mechanism dynamically routes each sample to an optimal combination of tokens. The reviewers all plead for the acceptance of the paper, albeit weakly. Among them, there is agreement that a timely problem is being tackled, that the paper is technically sound, and that convincing gains are achieved on relevant DGOD benchmarks. The reviewers highlighted the method's good performance, the careful way in which ablations of the three main components were applied, and the relevance of style-aware adaptation in the frozen-VFM setting. The main concerns were about limited originality, the claim that style induces style-dependent task mismatches, the questionnable style representation quality and the computational overhead. The rebuttal addressed these concerns well and the least positive reviewer raised their score as a result. That lead to the consensus of 'weak accept', which to follow, I feel, is justified. For the camera-ready version, I encourage the authors to further sharpen the conceptual positioning and add the extra analyses promised in the rebuttal.